# Diatomite derived hierarchical hybrid anode for high performance all-solid-state lithium metal batteries

Fei Zhou [1], Zheng Li[2], Yu-Yang Lu[3], Bao Shen[1], Yong Guan[4], Xiu-Xia Wang[5], Yi-Chen Yin[1], Bai-Sheng Zhu[1], Lei-Lei Lu[1], Yong Ni [3], Yi Cui [6], Hong-Bin Yao[1] & Shu-Hong Yu [1,4]

Lithium metal based anode with hierarchical structure to enable high rate capability, volume change accommodation, and dendritic suppression is highly desirable for all-solid-state lithium metal battery. However, the fabrication of hierarchical lithium metal based anode is challenging due to the volatility of lithium. Here, we report that natural diatomite can act as an excellent template for constructing hierarchical silicon-lithium based hybrid anode for high performance all-solid-state lithium metal battery. This hybrid anode exhibits stable lithium stripping/plating performance over 1000 h with average overpotential lower than 100 mV without any short circuit. Moreover, all-solid-state full cell using this lithium metal composite anode to couple with lithium iron phosphate cathode shows excellent cycling stability (0.04% capacity decay rate for 500 cycles at 0.5C) and high rate capability (65 mAh $g^{-1}$ at 5C). The present natural diatomite derived hybrid anode could further promote the fabrication of high performance all-solid-state lithium batteries from sustainable natural resources.

[1] Division of Nanomaterials & Chemistry, Hefei National Laboratory for Physical Sciences at the Microscale, CAS Center for Excellence in Nanoscience, Hefei Science Center of CAS, Department of Chemistry, University of Science and Technology of China, 230026 Hefei, Anhui, China. [2] Department of Polymer Science and Engineering, University of Science and Technology of China, 230026 Hefei, Anhui, China. [3] CAS Key Laboratory of Mechanical Behavior and Design of Materials, Department of Modern Mechanics, University of Science and Technology of China, 230026 Hefei, Anhui, China. [4] National Synchrotron Radiation Laboratory, University of Science and Technology of China, 230026 Hefei, Anhui, China. [5] Center for Micro- and Nanoscale Research and Fabrication, University of Science and Technology of China, 230026 Hefei, Anhui, China. [6] Department of Materials Science and Engineering, Stanford University, Stanford, CA 94305, USA. Correspondence and requests for materials should be addressed to H.-B.Y. (email: yhb@ustc.edu.cn) or to S.-H.Y. (email: shyu@ustc.edu.cn)

Lithium (Li) metal anode is one of the most attractive anodes for next-generation rechargeable batteries, due to its lowest electrochemical potential (−3.04 V vs. standard hydrogen electrode) and highest theoretical specific capacity (3860 mAh g$^{-1}$)[1–3]. However, the application of Li metal anode in conventional liquid electrolyte-based batteries has been long-term hindered by safety issues caused by thermodynamic instability of Li metal and the flammability of organic liquid electrolyte. The heterogeneous and fragile solid electrolyte interphase (SEI) caused by the high reactivity between Li metal anode and liquid electrolyte raises up the growth of dendrite, resulting in the short circuit, thermal runaway, and eventually uncontrollable safety issues[4–9]. Although much effort has been made recently to reinforce as-formed SEI layer or build up an artificial SEI layer to stabilize Li metal anode to some extent, the flammable nature of liquid organic electrolyte still makes it almost unacceptable in Li metal-based batteries[10,11].

Replacement of liquid electrolytes by uninflammable solid electrolytes (SEs) to build up all-solid-state Li metal batteries is in high demand for future high-energy density and safe energy-storage systems[12–15]. Generally, the SEs could be divided into inorganic solid electrolytes (ISEs) and solid polymer electrolytes (SPEs)[11,12]. Although ISEs usually exhibit high ionic conductivity at room temperature and good mechanical strength[16–19], the performances of solid-state Li metal batteries based on ISEs are limited by poor rigid solid–solid contact, leading to great interfacial resistance and low interfacial compatibility between ISEs and Li metal anode[20,21]. The uneven current density distribution at the interface of Li metal and ISEs would further promote the dendrite growth along the grain boundaries of ISEs[22,23]. In contrast, the SPEs comprising a soft polymer matrix and a Li ion salt exhibit much better flexibility and wettability with Li metal anode than ISEs, endowing intimate interface contact and lower interfacial resistance[24–27]. However, the low Li ion conductivities of SPEs at crystalline states, in particular, polyethylene oxide-based solid polymer electrolytes (PEO-SPEs), require the operating temperature over glass-phase transition temperature[25,28], with sacrificing the mechanical strength to suppress the growth of Li dendrites[22,26,29].

Considering beyond the electrolytes, the aforementioned limitations of SEs in solid-state Li metal batteries are significantly originated from the limited active surface area and low volume accommodation capability of mostly used planar Li metal anodes. A hierarchically structured design in advanced Li metal anode to improve its rate capability, accommodate the volume change, and eliminate dendrite growth is highly demanded in solid-state Li metal batteries[14,30–35]. It can be expected that a suitable combination of hierarchically structured Li composite anode with PEO-SPE could have the following advantages: the intensively enhanced electrode–electrolyte contact for low interfacial resistance and therefore improving the rate capability; the hierarchical ionic conductive framework to enable homogeneously stripping/plating of Li and maintain the integrity of the whole electrode.

To realize the merits of a hierarchically structured Li composite anode with PEO-SPE, a unique hierarchical framework material has to reach the requirements of low cost, good rigidity, high affinity to Li, and enough pore spaces[30,31,34] to accommodate sufficient Li inside. Nature is a master to produce various hierarchical materials and has offered us many inspirations for advanced battery electrode designs[36–38]. For instance, a pomegranate-inspired nanoscale design has been used to accommodate large volume change of silicon anode and achieve high areal capacity[36]. After screening various hierarchically structured materials from nature, we find that the diatomite is an ideal hierarchical framework material to meet the requirements for preparing hierarchical Li composite anode. First,

naturally abundant diatomite is low cost and its unique hierarchical structure with extremely high porosity is attractive for Li accommodation[39–41]. Second, with a facile magnesium reduction treatment, the unique hierarchical structure of diatomite could be totally inherited into the as-obtained silicon framework[42]. Third, the highly lithiophilic property of hierarchical silicon framework facilitates the lithiation process to yield a rigid Li$_{4.4}$Si framework to loading Li metal inside, forming a unique lithium silicide–Li hybrid anode[39,43].

Herein, we report a stable, dendritic free, and hierarchically structured Li metal-based hybrid anode derived from natural diatomite to realize high-performance all-solid-state Li metal battery. The natural diatomite is first transformed into a hierarchical silicon framework via a magnesiothermic reduction process. As shown in Fig. 1a, the diatomite-derived silicon (DF-Si) powder is mixed with molten Li to yield hierarchical lithium silicide–Li hybrid fragments, which could be cold-pressed into a composite anode after the PEO-SPE coating decoration. In as-fabricated PEO-SPE-coated diatomite-derived lithium silicide–Li (PEO-DLSL) composite anode, Li is embedded in the hierarchical pores of the Li$_{4.4}$Si framework and PEO coatings, resulting in a high electroactive contact area to homogenize the Li$^+$ ion flux and enable the anode integrity without Li dendrite growth (Fig. 2b). In contrast, the limited electroactive area of a planar Li-foil anode cannot efficiently avoid the growth of Li dendrites, which would easily penetrate the soft PEO-SPE causing the short circuit.

## Results

**Diatomite-templated fabrication of a hierarchical silicon framework for lithiation.** An individual diatomite has a Petri dish morphology with a fine hierarchical pore structure, which is confirmed by both the scanning electron microscopy (SEM) and transmission electron microscopy (TEM) images (Fig. 2a, d). The bigger-sized pores (400–800 nm in diameter) are distributed in the central area, whereas the smaller-sized pores (100–200 nm in diameter) are uniformly distributed in the whole diatomite framework (DF). After a facile magnesiothermic reduction process, the main phase of the diatomite framework could be converted into SiO (denoted as DF-SiO) or Si (denoted as DF-Si) from initial SiO$_2$ (JCPDS 46-1045), depending on the mass ratio of Mg powder to pristine diatomite. According to the powder X-ray diffraction (PXRD) analysis (Supplementary Fig. 1), the SiO phase (JCPDS 30-1127) could be obtained if the mass ratio of Mg to DF is 0.5:1. When the mass ratio increased to 1:1, the pure phase Si (JCPDS 27-1402) could be yielded. In addition, the PXRD peaks corresponding to the crystalline phase SiO$_2$ in diatomite (asterisk) gradually disappeared with increasing the amount of Mg powder. As confirmed by the SEM and TEM images (Fig. 2b, c and e, f), the generated DF-SiO and DF-Si have a similar hierarchical structure as that of pristine diatomite, especially in which the pores are maintained very well. To further reveal a good hierarchical pore structure in as-obtained frameworks, we conducted nitrogen adsorption/desorption analysis to study the variation of the specific surface area and pore structures of pristine DF, DF-SiO, and DF-Si, respectively (Fig. 2g–i). The surface area of the DF-SiO and DF-Si analyzed by Brunauer–Emmett–Teller (BET) method was 199.6 m$^2$ g$^{-1}$ and 271.6 m$^2$ g$^{-1}$, respectively, higher than that of pristine DF (100.3 m$^2$ g$^{-1}$), indicating that the removing oxygen by magnesiothermic reduction process produced more void space in the framework. The three-dimensional (3D) structures of pristine DF, DF-SiO, and DF-Si were further characterized by the soft X-ray tomography (SXT) as well (Fig. 2j–l, Supplementary Movies 1–3), in which the translucent pink area is corresponding to the solid

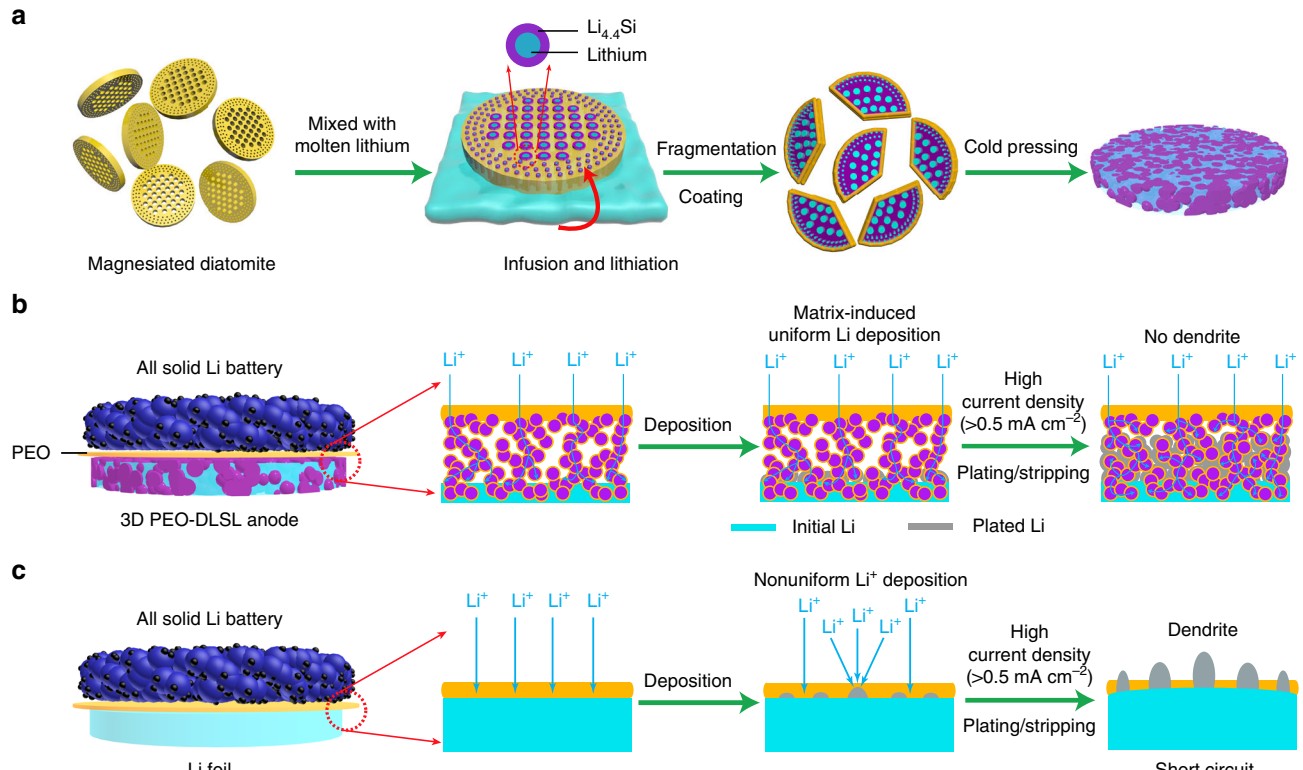

**Fig. 1** Schematic illustration of the PEO-DLSL anode fabrication process and Li stripping/plating behaviors in comparison with a planar Li anode. **a** The fabrication process of PEO-DLSL. The DF-Si powder was mixed with molten Li to form the DLSL fragments and the PEO-DLSL composite anode was fabricated by cold-pressing the DLSL fragments with PEO-SPE coatings. **b, c** Li stripping/plating behavior in all-solid-state Li metal batteries with PEO-DLSL anode and a planar Li-foil anode, respectively. The effective Li$^+$ ion flux in PEO-DLSL anode was much more uniform than that in a planar Li foil, leading to a dendrite-free anode without the short circuit

Si-based component, while the blue area could be ascribed to the hierarchical pores in the framework. It is evident that DF-Si behaves as a more porous structure with smaller sizes than that of pristine DF and DF-SiO, which will facilitate the lithiation process of as-obtained frameworks.

To evaluate the potentials of DF to fabricate a hierarchical hybrid anode via direct lithiation process, the aforementioned three products were mixed with molten Li with the same mass ratio of $M_{DF/DF-SiO/DF-Si}:M_{Li} = 1:1.6$ to proceed the lithiation of frameworks. Then the obtained lithiated powders were cold-pressed into pallets, as the electrodes and their Li-stripping performances were tested and compared (Supplementary Fig. 2, stripped to 1.0 V vs. Li$^+$/Li). For the lithiated DF-Si anode (DF-Si–Li), a total specific capacity of ~1153 mAh g$^{-1}$ could be extracted (based on the total electrode weight), in which the specific capacity of ~838 mAh g$^{-1}$ below the potential of 0.2 V is attributed from the Li metal. In contrast, the extracted specific capacity of DF-SiO-Li and DF-Li electrode was ~1031 mAh g$^{-1}$ and ~924 mAh g$^{-1}$, with only ~399 mAh g$^{-1}$ and ~265 mAh g$^{-1}$ of capacity contributed from the stripping of Li metal rather than the delithiation from Li$_{4.4}$Si or Li$_2$O, respectively. This is because the DF-Si consumed a least amount of Li to achieve full lithiation without formation of Li$_2$O, leaving a maximum amount of Li to serve as Li metal anode. Beyond that, the lithiation product of DF-Si is pure Li$_{4.4}$Si instead of the Li$_{4.4}$Si-Li$_2$O composites formed in DF-SiO-Li and DF-Li framework, which further lowers the inner resistance of the as-formed hybrid anode. As a result, the interfacial resistance of DF-Si–Li is superior to DF-SiO-Li and DF-Li[44,45]. Therefore, it is evident that the full magnesiothermic reduction of diatomite into a silicon framework is required for a desirable hybrid anode framework fabrication.

**Overstoichiometric lithiation of DF-Si to prepare a hybrid anode.** To study the influence of the lithiation extent on DF-Si, a different amount of Li (0.2, 0.5,, and 0.8 g) was employed to react with the same mass (0.5 g) of DF-Si powder. The obtained products were denoted as DF-Si-Li$_{0.2}$, DF-Si-Li$_{0.5}$, and DF-Si-Li$_{0.8}$, respectively. When the amount of Li was 0.2 g, the fractured fragments from the DF disk (Fig. 3a) could be observed due to the strain and volume expansion induced by the lithiation (Fig. 3b). When the amount of Li increased to 0.5 g, microparticles of lithiated DF were formed (Fig. 3c). By further increasing the amount of Li to 0.8 g, the dense microparticles were generated, which could be ascribed to the embedding of Li into the hierarchical Li$_{4.4}$Si particles (Fig. 3d). The photograph of the products at the different extents of lithiation (Supplementary Fig. 3) also showed the overstoichiometric lithiation process. The color of products changed from brown (similar to that of DF-Si) to black (the color of Li$_{4.4}$Si) and then to silvery gray (the mixed color of black Li$_{4.4}$Si and glossy Li). To further reveal the peculiarity of a hierarchical structure of DF-Si for promoting the embedment of Li, we applied a "phase field" method[46,47] (Supplementary Note 1 and Supplementary Table 1) to simulate the variation of Li concentration and stress evolution in the DF-Si framework during the lithiation process. As shown in Fig. 3e, the Li concentration in the DF-Si framework first increases at the edge of macropores and then spreads out to the whole framework, coinciding with the lithiation initiation from the macropore surface contacted with molten Li and the followed migration of the Li front toward the unlithiated silicon framework. This indicates that the hierarchical structure of DF-Si played a crucial role on the thoroughly gradual spreading of Li in the matrix. Also, the lithiation-induced expansion of the as-formed lithiated phase

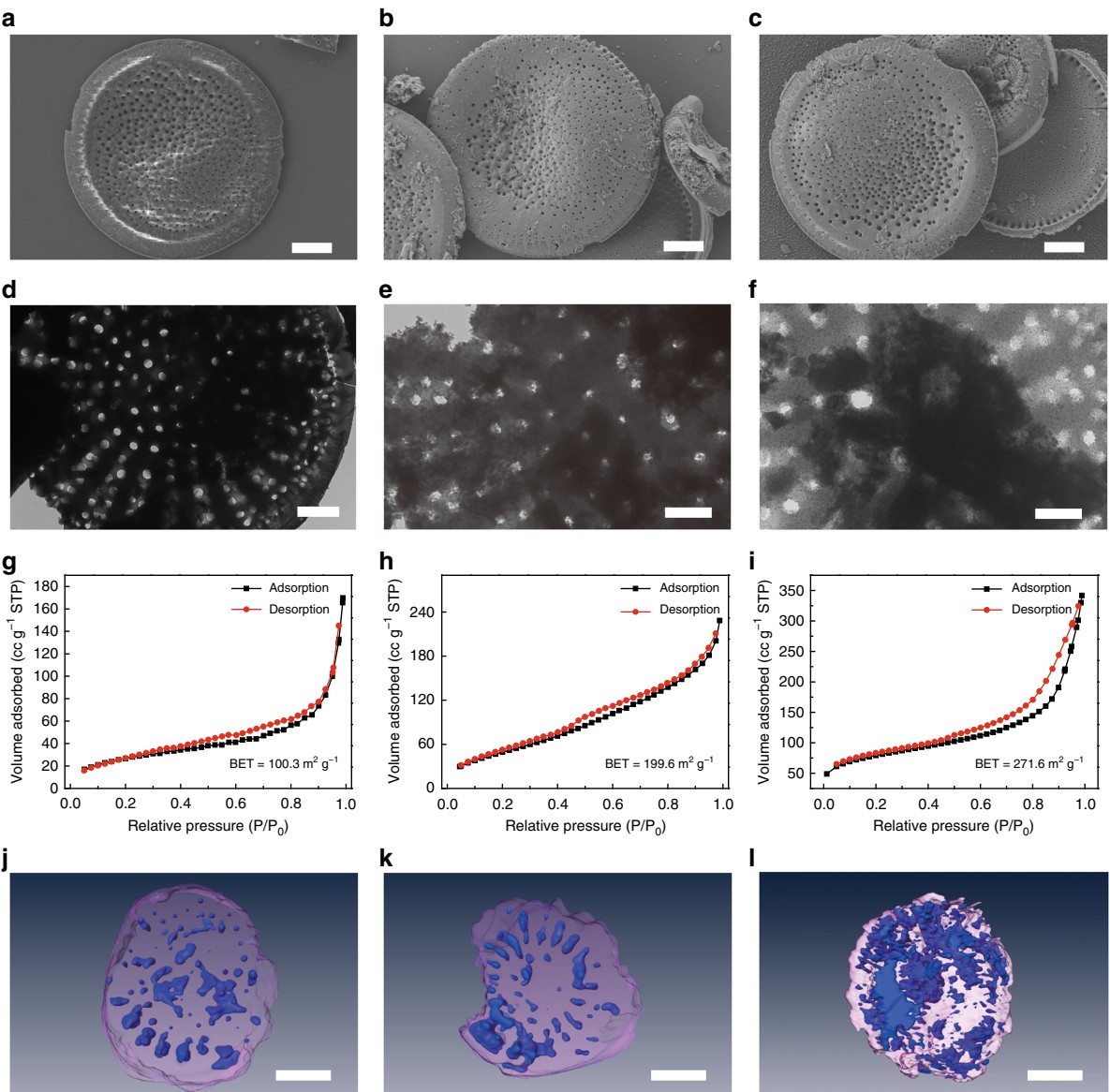

**Fig. 2** Morphology and surface area characterizations of pristine DF, DF-SiO, and DF-Si, respectively. **a–c** SEM images of pristine DF (**a** scale bar = 5 μm), DF-SiO (**b** scale bar = 5 μm), and DF-Si (**c** scale bar = 5 μm), respectively. **d–f** TEM images of pristine DF (**d** scale bar = 1 μm), DF-SiO (**e** scale bar = 500 nm), and DF-Si (**f** scale bar = 500 nm), respectively. **g–i** N$_2$ adsorption/desorption curves of pristine DF (**g**), DF-SiO (**h**), and DF-Si (**i**), respectively. **j–l** SXT images of DF (**j** scale bar = 2 μm), DF-SiO (**k** scale bar = 2 μm), and DF-Si (**l** scale bar = 2 μm), respectively

stretches the unlithiated silicon (Supplementary Fig. 4). As a consequence, the unlithiated part suffers from tensile stress, which promotes the propagation of cracks and leads to the fragmentation of lithiated DF-Si. Comparably, the lithiation simulation of silicon microflake without hierarchical pores showed a totally different Li concentration and stress evolution within the lithiation process. In the dense silicon microflake, the lithiation preferred to occur at the edge of silicon microflake and only slowly spread from the edge to the center area (Supplementary Fig. 5). Due to the absence of macropores, the stress amplitude in the dense silicon microflake is lower than that in the DF-Si framework. Besides the simulation, a control lithiation experiment was conducted on silicon microparticle powder to show the superior Li uptaking capability of DF-Si. As shown in Supplementary Fig. 6, after reaction with 0.8 g of Li, the silicon microparticle first became a lump of black mixture tightly adhered onto the bottom of a tantalum crucible after the lithiation, strongly contrasting to the silvery powder of as-formed

DF-Si-Li$_{0.8}$. To confirm the loading of Li in the hierarchical particles, the DF-Si-Li$_{0.8}$ microparticles were treated by focused ion beam (FIB) etching (Supplementary Fig. 7), which showed the appearance of interconnected pores in the matrix of DF-Si-Li$_{0.8}$, indicating the embedment of Li in the as-formed lithium silicide framework. The SXT image of an individual DF-Si-Li$_{0.8}$ microparticle (Fig. 3f, Supplementary Movie 4) was rendered according to X-ray absorption differences of Li and Li$_{4.4}$Si (Supplementary Fig. 8). We could also observe the embedment of Li (red area) in the interconnected pores of the Li$_{4.4}$Si framework (translucent purple area), forming the hierarchically structured hybrid anode. This hybrid anode is highly desirable to endow the good Li distribution in a sub-microscale hierarchical framework, leading to more uniform Li$^+$ flux and faster Li$^+$ transfer rate of solid Li metal anode.

The phase change of the lithiated DF-Si with using different amounts of Li was revealed by the PXRD (Fig. 3g). The main phase of DF-Si-Li$_{0.2}$ could be still indexed as silicon (JCPDS

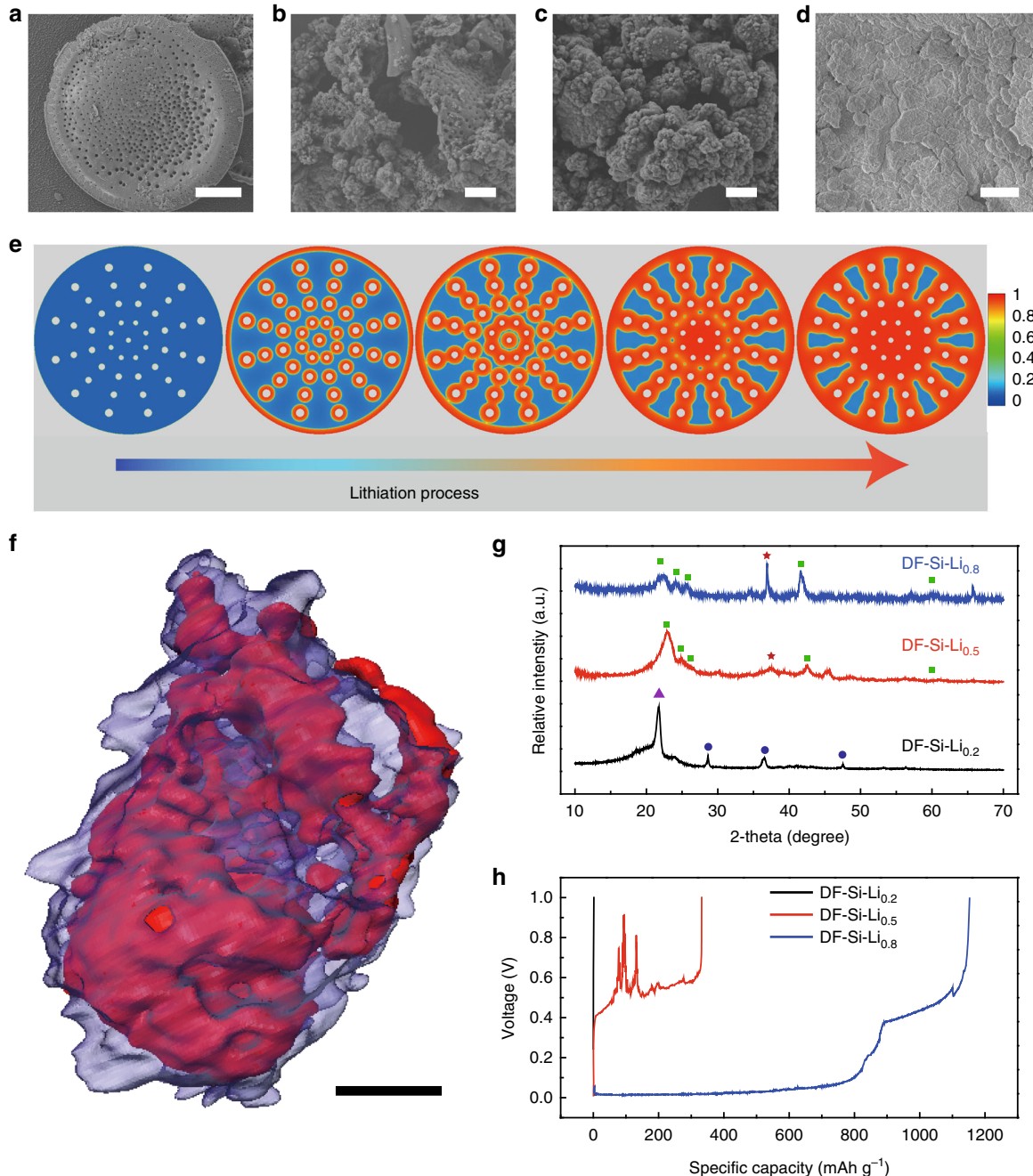

**Fig. 3** Characterizations and simulations of DF-Si powder lithiated by different amounts of Li. **a–d** SEM images of pristine DF-Si, DF-Si-Li$_{0.2}$, DF-Si-Li$_{0.5}$, and DF-Si-Li$_{0.8}$, respectively (scale bar = 10 μm for Figure a, scale bar = 5 μm for Figure **b**, **c**, **d**). **e** The simulation of Li diffusion into the DF-Si framework during the lithiation process. The variation from the blue color to red color represents the increase of Li concentration and lithiation extent of DF-Si along with the lithiation process. **f** SXT image of the DF-Si-Li$_{0.8}$ microparticle (scale bar = 1 μm). The image was rendered based on the difference of X-ray adsorption. The red color represents Li$_{4.4}$Si, while blue color corresponds to Li. **g** PXRD patterns of DF-Si lithiated with different amounts of Li. With a higher Li/DF-Si mass ratio, the relative intensity of the metallic peak (red asterisk) became more significant. The blue triangle denotes the peak of sample pack, the blue cycles denote the peaks of Si, and the green squares denote the peaks of Li$_{4.4}$Si. **h** Li- stripping voltage plateau curves of different DF-Si–Li anodes

No. 27-1402), demonstrating that only a little amount of DF-Si was lithiated at this low amount of Li. For DF-Si-Li$_{0.5}$, the main phase could be ascribed to Li$_{4.4}$Si (JCPDS No. 18-0747) but the metallic Li (asterisk, JCPDS 65-9346) also existed due to the accommodation of a hierarchical framework for Li. When the mass of Li further increased to 0.8 g, the relative intensity of metallic Li peaks became more significant, indicating that the overstoichiometric lithiation was conducted and the Li was accommodated in the matrix of Li$_{4.4}$Si as confirmed by SXT

(Fig. 3f). To demonstrate the essential role of the overstoichiometric amount of Li, we further tested the specific capacities that can be extracted from different lithiated DF-Si products. As shown in Fig. 3h, the DF-Si-Li$_{0.8}$ anode showed a much lower charge overpotential and a higher specific capacity compared with the DF-Si-Li$_{0.5}$ and DF-Si-Li$_{0.2}$. In particular, the main specific capacity contribution of DF-Si-Li$_{0.8}$ was from the metallic Li stripping (838 mAh/g, charge potential < 0.2 V), while the specific capacity of DF-Si-Li$_{0.5}$ was all from the delithiation of

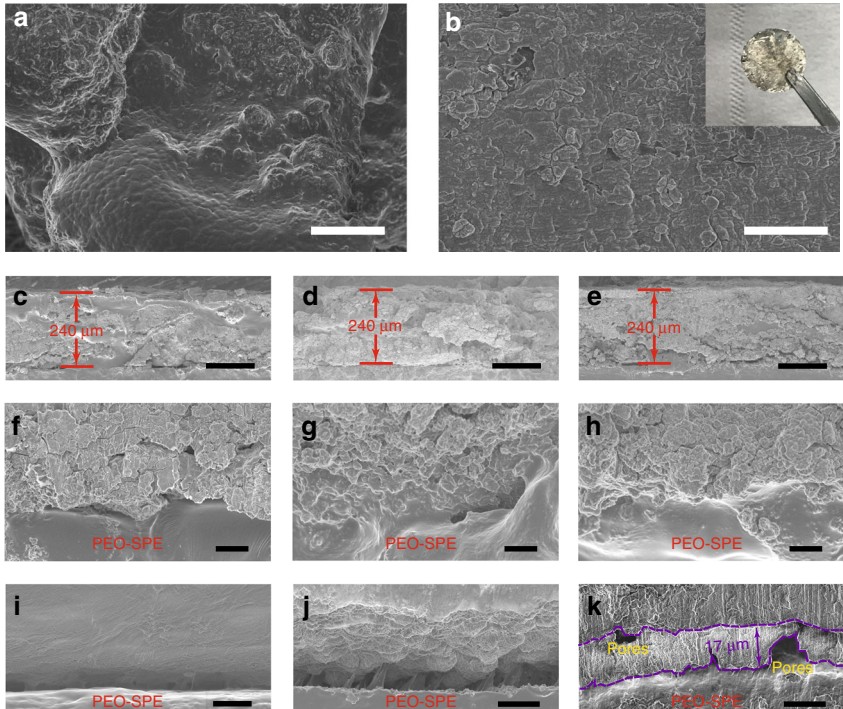

**Fig. 4** Morphological and structural characterizations of PEO-DLSL and PEO-Li-foil electrode at different states. **a** SEM image showing the DF-Si-Li$_{0.8}$ microparticles after a thin layer of PEO-SPE coating (scale bar = 50 μm). **b** SEM image demonstrating the flat surface morphology of the PEO-DLSL electrode (scale bar = 20 μm). The inset is the photograph of the as-fabricated composite anode. **c–e** Cross-sectional SEM images of pristine PEO-DLSL electrode (**c**), after 4 mAh cm$^{-2}$ of Li stripping (**d**), and 4 mAh cm$^{-2}$ of plating Li back (**e**). The current density was set at 0.5 mA cm$^{-2}$ for all of the above characterizations (scale bar = 200 μm). **f–h** The corresponding magnified SEM images of the electrode from **c** to **e**, showing the morphologies at the interface of the PEO-DLSL electrode and PEO-SPE (scale bar = 5 μm). **i–k**, Cross-sectional SEM images showing the morphologies at the interfaces of Li foil and PEO-SPE (scale bar = 20 μm). The intimal surface of Li foil is smooth (**i**). After 4 mAh cm$^{-2}$ of Li stripping out, a rough Li-foil surface could be observed (**j**). When 4 mAh cm$^{-2}$ of Li was plated back, a layer of deposited Li (~17 μm) appeared at the interface with pores and cracks (**k**)

Li$_{4.4}$Si (332 mAh/g at the potential of 0.4~1.0 V). The charge curve of DF-Si-Li$_{0.8}$ is very smooth, whereas DF-Si-Li$_{0.5}$ is quite fluctuant due to the instability of Li$_{4.4}$Si frameworks during the Li stripping. The DF-Si-Li$_{0.2}$ electrode showed minimal specific capacity of only 6.4 mAh/g and the voltage raised very fast over 1.0 V. Therefore, the overstoichiometric Li is very crucial to achieve desired electrochemical behaviors of Li metal anode. In this case, the DF-Si-Li$_{0.8}$ hybrid anode materials were used to fabricate all-solid-state Li metal composite anodes.

**Fabrication and characterizations of PEO-DLSL composite anode.** To fabricate the PEO-DLSL composite anode, a thin layer of PEO-SPE was first coated onto the surface of the DF-Si-Li$_{0.8}$ microparticles by a solution mixing and drying process (see the "Methods" section). As shown in Fig. 4a, the surface of DF-Si-Li$_{0.8}$ microparticles became smooth and no cracks could be observed, implying the uniform coating of PEO-SPE. After facile cold-pressing of as-synthesized powders, the obtained PEO-DLSL electrode shows silvery color on the surface (inset in Fig. 4b), and the SEM image of the electrode surface indicates its uniform and dense structure (Fig. 4b). The magnified cross-sectional SEM image at a stretched point (Supplementary Fig. 9a) clearly shows the silk-like structure of an elastic PEO-SPE to connect the stretched solid electrode. Furthermore, the energy-dispersive X-ray spectroscopy (EDX) mapping results (Supplementary Fig. 9b, c) show that the oxygen signal (blue) well overlaps with the PEO-SPE coating and the silicon signal (green) distributes throughout the solid electrode, implying the uniform PEO-SPE coating onto the whole 3D network of the DF-Si-Li$_{0.8}$ matrix. In this composite anode, the elastic PEO-SPE could enhance the contact area

between the hybrid anode and solid electrolyte. Further, PEO-SPE could construct a continuous Li$^+$ ion transport channel throughout the entire electrode, thus shortening Li$^+$ ion transportation pathway and homogenizing the Li$^+$ ion flux in the solid composite electrode. It is also worth noting that the PEO-SPE exhibits a relatively high Li$^+$ ion conductivity at the operating temperature (4.63 × 10$^{-4}$ S cm$^{-1}$ at 60 °C, Supplementary Fig. 10). With the assistance of PEO-SPE, Li$^+$ ion diffusion and transportation capability in the PEO-SPE/Li–Si composite framework could be significantly improved. Therefore, the PEO-DLSL electrode could have a stable electrode dimensionality during the Li stripping/plating, endowed by both the rigid 3D Li$^+$ ion-conductive framework of lithiated silicon and PEO-SPE coatings. To show the sturdiness of the as-constructed 3D electrode network at the solid state, the PEO-DLSL electrode in a half-cell was charged to 1.0 V to ensure fully tripping out of all Li. The obtained delithiated electrode still kept good rigid structural integrity, with only the color change from silvery to black, because of the formation of a porous framework, indicating the stability and interconnected nature of the PEO-DLSL framework (Supplementary Fig. 11a). In addition, hierarchical pores could be observed from the SEM image of the "black" area of the stripped electrode, where it was initially occupied by Li (Supplementary Fig. 11b).

Furthermore, the ex situ SEM characterizations were carried out to show the thickness variation of the electrode and the interfacial stability between the electrode and PEO-SPE in one cycle of the Li stripping/plating. As shown in Fig. 4c–e, the thickness of the PEO-DLSL electrode was maintained around ~240 μm during the stripping out and plating back 4 mAh cm$^{-2}$

of Li, indicating that the Li stripping and plating occurred inside the 3D framework of the composite electrode. The corresponding magnified SEM images (Fig. 4f–h) show the uniform electrode structure and dense interface between the electrode and PEO-SPE, further confirming the homogeneous Li stripping/plating in the network of the composite electrode. In contrast, for a planar Li-foil anode, after stripping out 4 mAh cm$^{-2}$ of Li, the surface of Li foil at the electrode/electrolyte interface became from smooth (Fig. 4i) to uneven like hills (Fig. 4j). When 4 mAh cm$^{-2}$ of Li was plated back, an independent layer of Li could be observed at the PEO-SPE/Li-foil interface. The average thickness of the deposited Li was about ~17 μm, which was in consistent with the amount of plated Li (Fig. 4k). Besides, the pores and cracks appeared at both interfaces of the deposited Li layer, indicating the reduced contact area and potential growth of dendrites. It is predictive that the contact issues of Li foil became more severe with repeat Li stripping/plating process due to the extensive volume change and uneven Li deposition behavior.

**Electrochemical performance of the PEO-DLSL anode in comparison with the Li-foil anode**. To demonstrate the good Li$^+$ ion transfer property of the PEO-DLSL composite anode, the interfacial resistance of the symmetric cell was evaluated at first by electrochemical impedance spectroscopy (EIS). According to the EIS results in Supplementary Fig. 12, the interfacial resistance of the PEO-DLSL cell was only 124 Ω cm$^{-2}$, while the value of Li-foil cell was 397 Ω cm$^{-2}$, confirming the improved interfacial Li$^+$ transportation induced by the 3D Li$^+$ ion-conductive framework in the composite anode. As shown in Fig. 5a, the PEO-DLSL symmetric cell exhibited a smaller Li stripping/plating overpotential compared with the Li-foil cell at 60 °C. When the current density is 0.1 mA cm$^{-2}$, the average overpotential for the PEO-DLSL cell was 18 mV, lower than that of Li-foil cell (38 mV, Fig. 5b). The difference became more significant when the cell operated at a higher current density. For example, when the current density was 0.5 mA cm$^{-2}$, the average overpotential of PEO-DLSL was only 98 mV, while the value of Li foil was 276 mV (Fig. 5c). It was observed that the voltages slowly decreased to zero when the polarizations were switched at the current density higher than 0.2 mA cm$^{-2}$. This could be attributed to the concentration polarization (CP) when cycled at high current density. Li$^+$ ion transportation and diffusion rate at the electrode/electrolyte interface and in the electrolyte is order of magnitudes smaller than the electrochemical reaction rate. Therefore, the depolarization of CP required time to recover homogeneous Li$^+$ ion concentration distribution and achieve a steady state, due to the unsatisfied Li$^+$ ion conductivity in the solid-state cell, and thus the voltages slowly decreased to zero when the polarizations were switched off. Furthermore, the Li$^+$ ion diffusion pathway in our 3D hierarchical PEO-DLSL electrode is much longer than that at the planar Li-PEO electrolyte interface. Hence, the slowly decreasing of voltage is more obvious for PEO-DLSL than Li foil. Noticeably, at the current density of 1 mA cm$^{-2}$, the overpotential of PEO-DLSL was 269 mV, while the Li foil cannot be operated at 1 mA cm$^{-2}$ due to the large overpotential exceeding the instrument voltage range (5 V). The PEO-DLSL electrode can be operated at 2 mA cm$^{-2}$ as well, which exhibited stable cycling performance over 95 h with polarization voltage around 4 V (Supplementary Fig. 13). Moreover, the PEO-DLSL anode outperformed the planar Li foil in terms of cycling stability (Fig. 5d). More than 1000 h of stable Li stripping/plating with little overpotential increase can be realized at a relatively high current density and areal capacity (0.5 and 0.5 mAh cm$^{-2}$). In contrast, the planar Li-foil cell showed a gradual increase in voltage hysteresis over cycles, due to the accumulating interfacial impedance,

followed by an internal short circuit within 100 h. The superior rate and cycling performance of PEO-DLSL undoubtedly demonstrate the key role of a 3D hierarchical structure derived from diatomite on improving the performance of solid composite anode.

To demonstrate the feasibility of a hierarchal PEO-DLSL composite anode for all-solid-state Li battery, the full cells coupled with LiFePO$_4$ (LFP) cathodes and PEO-SPE layers were assembled to test the rate capability and long-term cycling stability. Supplementary Fig. 14a shows the EIS results of full cells based on the PEO-DLSL anode and Li-foil anode. The areal resistance of LFP/PEO-DLSL was ~90 Ω cm$^{-2}$, much smaller than that of LFP/Li foil (~190 Ω cm$^{-2}$), indicating the lower interfacial resistance in the PEO-DLSL-based full cell. As shown in Fig. 5e, the LFP/PEO-DLSL full cell displayed a much better rate performance compared with the Li-foil-based full cell. At low current densities, the specific discharge capacity of a planar Li-foil-based cell was 148 and 144 mAh g$^{-1}$ at 0.1 and 0.2 C (1 C = 170 mA g$^{-1}$, corresponding to 0.4 mA cm$^{-2}$), respectively, whereas the values of the PEO-DLSL-based full cell were 167 and 155 mAh g$^{-1}$. The Li-foil-based cell exhibited a very low specific charge capacity of only 7 mAh g$^{-1}$ at 5 C, indicating poor rate capability. In contrast, the PEO-DLSL-based cell still retained 65 mAh g$^{-1}$ at 5 C. After decreasing the current density back to 0.2 C, the capacity of PEO-DLSL-based full cell quickly increased to 152 mAh g$^{-1}$, disclosing a fine capacity recovery. The improved rate performance could be attributed to the fast Li$^+$ ion transportation property at both the interface and inside the electrode induced by the PEO-SPE coating and 3D Li$^+$ ion-conductive Li$_{4.4}$Si framework. This benefit was further confirmed by the galvanostatic discharge–charge (GDC) voltage profiles. As shown in Supplementary Fig. 14b, the LFP/PEO-DLSL and LFP/Li cells both demonstrated typical LiFePO$_4$ charge/discharge voltage plateaus at 0.2 C. However, when cycled at 2 C, the LFP/PEO-DLSL cell exhibited a much lower charge–discharge overpotential and a higher specific capacity than that of LFP/Li (Fig. 5f). As to the long-term cycling performance, the LFP/PEO-DLSL cell delivered the specific discharge capacity of 117 mAh g$^{-1}$ over 500 cycles with a very slow degradation rate (0.04% per cycle) at 0.5 C, whereas the LFP/Li-foil full cell was short circuited at the 70th cycle (Fig. 5g). The stable composite anode integrity and intimate interface contact together helped to improve the cycling stability of the as-fabricated all-solid-state LFP/PEO-DLSL full cell. Among the electrochemical performance summarized in Supplementary Table 2, our developed PEO-DLSL composite anode exhibited an outstanding performance in overpotential, cycling stability, and rate capability in comparison with the previously reported Li metal anode and SPE designs for solid-state Li batteries.

## Discussion
In our proposed hierarchical PEO-DLSL electrode design, the Li$^+$ ion-conductive channels are provided by the PEO-SPE decorated Li$_{4.4}$Si (PEO-Li$_{4.4}$Si) frameworks. In this context, owing to the high Li$^+$ ion conductivity of the PEO-Li$_{4.4}$Si framework, our fabricated PEO-DLSL composite anode with embedded Li in a porous Li$_{4.4}$Si framework and PEO-SPE coatings exhibit the following advantages. First, the porous Li$_{4.4}$Si framework could act as the host to accommodate Li in the hierarchical pores, resulting in a robust electrode structure for electrochemical plating/stripping of Li. Second, the 3D Li$_{4.4}$Si framework together with PEO-SPE coatings possesses a large electrode–solid electrolyte interfacial surface area, which could reduce localized current density and suppress the dendritic growth for improving the interfacial stability. To confirm this merit, we investigated the interfacial

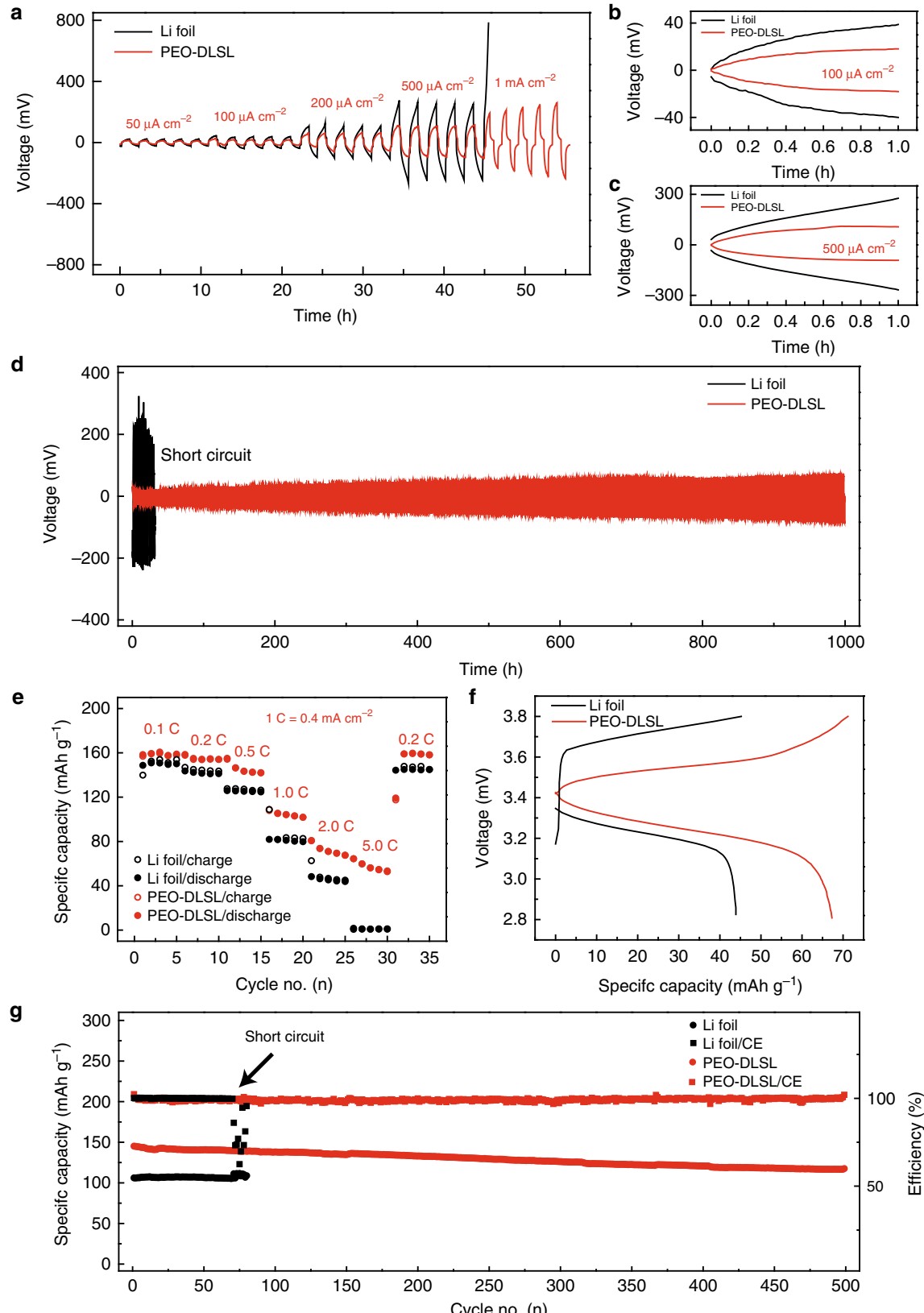

**Fig. 5** Electrochemical performance comparison of the cells using PEO-DLSL and planar Li-foil anode, respectively. **a** Voltage profiles of symmetric cells using different anodes at different current densities. **b**, **c** Detailed voltage profiles at the current density of 0.1 and 0.5 mA cm$^{-2}$, respectively. **d** Comparison of the long-term cycling stability of the symmetric cells with PEO-DLSL and Li-foil anodes at a current density of 0.5 mA cm$^{-2}$. **e** Rate capability comparison of Li-LFP full batteries with PEO-DLSL and Li-foil anode. **f** GDC curves of PEO-DLSL/LFP and Li-foil/LFP full cells at 2 C current density. **g** Long-term cycling performance of batteries at a current density of 0.5 C. All cells were operated at 60 °C

impedance variation of symmetric and full cells using PEO-DLSL and Li-foil anodes before and after cycling, and the results are shown in Supplementary Fig. 15. In the full cells, the original areal $R_{ct}$ of PEO-DLSL/LiFePO$_4$ was ~116 $\Omega$ cm$^{-2}$, and it changed a little to ~126 $\Omega$ cm$^{-2}$ after ten cycles. Significantly, the areal $R_{ct}$ of Li/LiFePO$_4$ increased obviously from ~153 to ~212 $\Omega$ cm$^{-2}$. For symmetric cells, the areal $R_{ct}$ of PEO-DLSL/PEO-DLSL and Li/Li was ~124 $\Omega$ cm$^{-2}$ and ~292 $\Omega$ cm$^{-2}$, respectively. After 20 cycles, the corresponding areal $R_{ct}$ was ~138 $\Omega$ cm$^{-2}$ and ~307 $\Omega$ cm$^{-2}$, respectively. These comparisons demonstrate that the significant role of the as-proposed 3D hierarchical design in improving the interfacial stability between the anode and the PEO-based solid electrolyte. Finally, the a 3D porous Li$_{4.4}$Si/PEO-SPE framework with a rigid mechanical characteristic could support the electrode with a negligible volume change during cycling, which would ensure intimate interfacial contact to further maintain the stability of the electrode.

To enable high efficiency of a hierarchical 3D electrode, the high Li$^+$ ion conductivity inside the electrode is highly desirable. In our fabricated composite anode, the Li$^+$ ionic conductivity is contributed both by the PEO-SPE decoration layer and Li$_{4.4}$Si framework. It is well known that PEO-SPE is a good Li$^+$ ion conductor when working at 60 °C, showing a high Li$^+$ ionic conductivity of $4.63 \times 10^{-4}$ S cm$^{-1}$ at 60 °C (Supplementary Fig. 10). This means that the Li$^+$ ions can be conducted along the surface of the Li$_{4.4}$Si framework in the as-fabricated hierarchical 3D composite anode. For the evaluation of Li$^+$ ionic conductivity in the Li$_{4.4}$Si framework, the previously reported Li$^+$ ion diffusion coefficient of Li$_{4.4}$Si was in the range from 10$^{-7}$ to 10$^{-13}$ cm$^2$ S$^{-1}$ at room temperature[48–53], making it hard to judge the capability of the Li$_{4.4}$Si framework for the Li$^+$ ionic conduction in our composite anode. In this case, we adopted the EIS comparison of the Li$_{4.4}$Si electrode and PEO-SPE decorated Li$_{4.4}$Si (PEO-Li$_{4.4}$Si) electrode to reveal the Li$^+$ ionic conduction contribution of the Li$_{4.4}$Si framework in our fabricated composite anode. As shown in Supplementary Fig. 16, the resistance of the Li$_{4.4}$Si electrode was measured as ~54 $\Omega$ cm$^{-2}$, which means that the Li$^+$ ions can be conducted in the Li$_{4.4}$Si framework. After making decoration of PEO on the framework, the measured resistance reduced a little to ~37 $\Omega$ cm$^{-2}$, indicating that the PEO decoration could enhance the Li$^+$ ion conductivity of the PEO-Li$_{4.4}$Si electrode. On the basis of the above results, we consider that the Li$^+$ ionic conduction in the as-fabricated hierarchical 3D composite anode is endowed by both the PEO-SPE decoration layer and the Li$_{4.4}$Si framework.

Beside the high Li$^+$ ion conductivity, the balance between the ionic conductivity and electronic conductivity in the solid Li metal anode is also needed for its interfacial stability. During the plating process, the Li would be deposited at where the sufficient Li$^+$ ions and electrons are both provided (electron/ion interface). In the traditional electrode, the electronic conductivity is much higher than Li$^+$ ion conductivity and thus the deposition of Li is likely to be appeared at the electrode/electrolyte interface. But in our fabricated composite electrode, the Li$^+$ ionic conductivity is dominant to make the deposition of Li occurring in the electrode framework, because of the low electronic conductivity of Li$_x$Si ($3.06 \times 10^4$ $\Omega$ cm)[54]. The electron transfer is likely to be conducted in the metallic Li. As demonstrated in Supplementary Fig. 17a, plating of Li within the PEO-Li$_{4.4}$Si framework starts from the bottom electron-conductive Li layer/PEO-Li$_{4.4}$Si interface (denoted as interface I). "Fresh" Li first arises at interface I and then grows along the horizontal direction to fill the gap (Supplementary Fig. 17b). At the same time, the deposited Li could conduct electrons to the upper adjacent framework particle surface, generating a new electron/Li$^+$ interface (interface II) and resulting in the rise of plated Li (Supplementary Fig. 17c). In our

opinion, Li plating occurs at two orientations to fill the whole framework up[31]. It is noteworthy that the difference in electronic transfer ability could also explain the low charge capacity and fluctuation curve of DF-Si-Li$_{0.2}$ and DF-Si-Li$_{0.5}$ electrodes.

In summary, we report an efficient strategy to construct 3D hierarchical PEO-DLSL composite anode derived from natural abundant diatomite. The hierarchical structure of the PEO-DLSL composite anode inherited from natural diatomite facilitates the intimate contact between the electrode and the solid-state electrolyte, providing highly Li$^+$ ion-conductive channels, which is crucial for accommodating the interfacial fluctuation during battery cycling. As a result, low overpotential and good cycling stability could be realized in symmetric and full cells using PEO-DLSL composite anodes. This hierarchical Li composite anode is promising to fabricate a safe solid-state Li metal battery with high energy/power density.

## Methods

**Purification of diatomite.** Ultrapure bio-silica microflakes (BSM) were obtained from raw diatomite after a specific purification process. Typically, the as-received diatomite powders were immersed and stirred overnight in sulfuric acid (1 M) and nitric acid (2 M) to remove the metal impurities and the organic constituents. The purified products were collected by filtration and washed by deionized water (DIW) and ethanol, respectively. Then the BSM with different particle sizes were separated using recycled sedimentation processes in acetone. Finally, the obtained microflakes were dried and annealed at 500 °C in air for 12 h.

**Magnesiothermic reduction of diatomite.** In total, 1.0 g of purified diatomite was mixed with 3.0 g of sodium chloride (Sinopharm Chemical Reagent Co., Ltd) and a different mass of magnesium powder (99.5%, Aladdin) by hand grinding. The mass of Mg was determined by the desired mass ratio ($M_{diatomite}/M_{Mg}$). Then the mixed powder was sealed in a tantalum crucible and transferred into a tube furnace. The crucible was then heated at 650 °C for 6 h in 5% H$_2$/Ar. The magnesiated diatomite (DF-SiO or DF-Si depending on the mass ratio) was dispersed in DIW and reacted with concentrated HCl and 0.5% HF (Sinopharm Chemical Reagent Co., Ltd) for 6 h by stirring. After that, the product was collected by filtration and washed with DIW and ethanol, respectively, and finally dried at 80 °C overnight.

**Overstoichiometric lithiation and PEO-SPE decoration of DF.** The synthesized DF powders were first heated at ~120 °C in the argon-filled glove box for 12 h to remove the adsorbed O$_2$/H$_2$O before performing overstoichiometric lithiation progress. To overlithiate the DF-Si, 0.5 g of DF-Si powders were put into a tantalum crucible and heated on a hot plate at ~350 °C. After that, Li metal (China Energy Lithium Co., Ltd) was weighted in a desired amount and put into the crucible under stirring to achieve a homogeneous reaction. Once the reaction was completed, the obtained products were cooled in the glove box to room temperature. PEO-SPE for surface decoration was synthesized by stirring a certain amount of PEO ($M_w = 100,000$, Alfa Aesar) and lithium bis(trifluoromethanesulfonyl)imide (LiTFSI, TCI Chemicals) in anhydrous acetonitrile (Energy Chemical, www.energy-chemical.com). Then the DF-Si–Li powders were mixed with PEO-SPE solution and stirred for 2 h, and then dried at 60 °C. The PEO-SPE used for the electrochemical test was fabricated by mixing a certain amount of PEO ($M_w = 600,000$, Sigma–Aldrich) and LiTFSI with gentle stirring to form a viscous solution, and then the solution was poured into a square Teflon mold (5 cm) and dried at 60 °C for 24 h. Finally, the PEO-SPE membrane was peeled off from the Teflon mold and cut into a round disk (diameter = 16 mm) for the electrochemical test.

**PEO-DLSL electrode fabrication.** Cold-pressing method was applied to fabricate the PEO-DLSL electrode. A certain amount of DF-Li-PEO powders were weighted and put into the pellet die (diameter = 12 mm) followed by ~10 tons of pressure. All the steps were conducted in the glove box with H$_2$O and O$_2$ content below 0.1 ppm. Typically, the weight of the PEO-DLSL was about ~60 mg.

**Li extraction behavior of different electrodes.** The Li extraction behavior of the synthesized electrodes was measured in 2032 coin cells and discharged to 1.0 V. The composite electrodes were used as a cathode, while the Li foil was used as an anode. The electrolyte used for Li extraction is 1 M LiPF$_6$ in EC/DEC (3:7, v:v) and the current density was 10 $\mu$A mg$^{-1}$.

**Electrochemical performance of symmetric cells.** Electrochemical performance of symmetric cells using PEO-DLSL and Li foil (diameter = 15.6 mm, www.dodochem.com) was evaluated by 2032-coin cell batteries. The cells were assembled in an argon-filled glove box with oxygen and water content below

0.5 ppm. The electrolyte was PEO-SPE ($M_w = 600{,}000$ for PEO with LiTFSI) for solid-state cells, while 1 M LiPF$_6$ in EC/DEC (www.dodochem.com) was used as liquid electrolyte. The plating/stripping performance of symmetric cells was conducted on a Land multichannel electrochemical testing system with different current density and areal capacity. Electrochemical impedance spectroscopy tests were recorded by a Bio-Logic VMP3 electrochemical working station between 1 M Hz and 1 Hz. It is noteworthy that we set the standing time as 30 min for the symmetric cell cycling when the current densities were higher than 200 µA cm$^{-2}$ to release possible polarization. As a result, the voltage slowly decreased to zero at the end of charge/discharge process.

**Electrochemical performance of all-solid-state LiFePO$_4$/Li batteries using different Li anodes.** To study the electrochemical performance of Li/LiFePO$_4$ batteries, 2032 coin cells were assembled. The anode was PEO-DLSL and planar Li foil, respectively. For preparing the working cathode, PEO-SPE was first made by mixing PEO and LiTFSI in anhydrous acetonitrile (Energy Chemical, www.energy-chemical.com) and used as a binder. Then, LiFePO$_4$ (MTI Inc.) powder, super-P acetylene black (Alfa Aesar), and PEO-SPE were mixed at a weight ratio of 80:10:10 in the N-methyl-2-pyrrolidone (NMP, Sigma-Aldrich) solvent to form a uniform slurry. Afterward, the slurry was casted on an Al-carbon foil (MTI) by a doctor blade and dried at 60 °C in vacuum. The diameter of the electrode slices was 11 mm with active material loading of ~2.4 mg cm$^{-2}$. The electrolyte was PEO-LiTFSI SPEs. The operating temperature was 60 °C. The rate capability test was carried out on a Land multichannel test system with different current densities between 2.3 and 3.8 V (vs. Li$^+$/Li).

**Characterizations**. Powder X-ray diffraction (PXRD) patterns were carried out on a Philips X'Pert PRO SUPER X-ray diffractometer equipped with graphite-monochromatized Cu Kα radiation. Transmission electron microscope (TEM, Hitachi H-7650) and scanning electron microscope (SEM, JEOL-6700F) were employed to visualize the morphologies, sizes, structures, and elemental compositions of the products. The nitrogen absorption/desorption isotherms were obtained at 77 K on a Quantachrome autosorb iQ2 automated gas sorption analyzer, using BET calculations for surface area and BJH for pore size distribution at 77 K. The soft X-ray beam was focused on the samples using an elliptical capillary condenser. A total of 122 projections were collected at tilt angles of −57 to 66° at 1° increments with 2 seconds exposures at 520 eV X-ray energy. All of the projections were corrected based on a reference image with a flat field intensity and aligned to the rotation axis. Tomographic reconstruction of the projections was carried out by the total variation (TV)-based simultaneous algebraic reconstruction technique.

## Data availability

The data that support the findings of this study are available from the corresponding author upon request.

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

## Acknowledgements

We acknowledge the funding support from the National Natural Science Foundation of China (Grants 51571184, 21501165, 21431006, 51571184, and 21805268), the Foundation for Innovative Research Groups of the National Natural Science Foundation of China (Grants 21521001 and 11472262), the Users with Excellence and Scientific Research Grant of Hefei Science Center of CAS (2015HSC-UE007, 2015SRG-HSC038), Key Research Program of Frontier Sciences, CAS (Grant QYZDJ-SSW-SLH036), the Chinese Academy of Sciences (Grant KJZD-EW-M01-1), the Fundamental Research Funds for the Central Universities (Grant WK 2060190085), and the Strategic Priority Research Program of the Chinese Academy of Sciences (Grant No. XDB22040502). This work was partially carried out at the USTC Center for Micro and Nanoscale Research and Fabrication.

## Author contributions

F.Z. and H.-B.Y. conceived the concept. H.-B.Y. and S.-H.Y. supervised the project. F.Z., Z.L. and B.S. carried out the synthesis and performed materials characterization and electrochemical measurements. Y.Y.L and Y.N. conducted the simulation of lithiation process and stress evolution. Y.G., X.-X.W. and B.-S.Z. assisted in the characterization of the electrode materials. Y.-C.Y. and L.-L.L. assisted in drawing figures and analyzing the electrochemical performance data. S.-H.Y. and Y.C. provided important experimental insights. F.Z., H.-B.Y. and S.-H.Y. cowrote the paper. All authors discussed the results and commented on the paper.

## Additional information

**Competing interests:** The authors declare no competing interests.

