## [Peer Review File · Nature Communications]

Reviewers' comments:

Reviewer #1 (Remarks to the Author):

This work introduces a new concept to design Li-metal based composite anode. The performance is good, and this work can be applied to current battery fabrication process. Here I have some questions to help authors further improve the quality of their work. My comments are given below:

1. The motivation of Li_{4.4}Si framework design is not clear. Please add more detailed discussion in the manuscript.
2. "In this composite anode, the elastic PEO-SPE could enhance the contact area between the hybrid anode and solid electrolyte, shortening ion transportation pathway and homogenizing the ion flux in the solid composite electrode." please specify the motivation of adding PEO-based electrolyte in the anode. What's the ionic conductivity for the composite anode and the lithium silicide-Li hybrid material?
3. Please explain why the hybrid anode is more stable than the pure Li metal anode in symmetric cell cycling. And authors compared the interfacial impedance in both anode, please specify what types of interfacial impedance are indicated in the manuscript.
4. In the all solid state battery, there is no electrolyte in the cathode, how to maintain proper ionic conductivity in cathode part in order to facilitate a full cell cycling?

Reviewer #2 (Remarks to the Author):

In this paper, the hierarchical silicon containing 3D anode was prepared and investigated in PEO based all-solid-state batteries. With this composite Si-Li alloy electrode, the anode/polymer-electrolyte interface was increased and therefore the area current density was lower, which are beneficial for homogenization of the Li⁺ flux across the interface and suppression of lithium dendrite at larger current density plating, two biggest challenges of the solid-state batteries. Higher current density performance verified the advantage of this specific anode design. In my opinion, this paper can be published after addressing following issues,

- (1) It has been reported that the flat lithium metal anode could also give stable long-term plating/stripping at lower current density, such as lower than 0.5mA/cm². The advantage of this 3D anode in solid-state cells will become significant in higher current density. A description of "higher current density plating/stripping" should be added in Figure 1b and 1c.
- (2) "the extracted specific capacity of DF-SiO-Li and DF-Li electrode was ~1031 mAh/g and ~924 mAh/g, with only ~399 mAh/g and ~265 mAh/g of capacity contributed from the Li, respectively" It is hard to understand that only 265 mAh/g of capacity contributed from the Li in the DF-Li, since the DF itself should not contribute obvious capacity. Please revise the description here and reduce the misunderstanding of the readers.
- (3) In Figure 3h, the DF-Si-Li alloys showed relatively low de-lithiation capacity compared with the previously reported SiLix alloys, such as the DF-Si-Li_{0.2} didn't give visible charge capacity when charge to 1.0V vs Li/Li⁺, the DF-Si-Li_{0.5} give a capacity of 332mAh/g with a voltage plateau at 0.4-0.5V without delithiation capacity between 0.1-0.4V. These features might because the lithium consumption in the formation of SEI in traditional liquid electrolyte. Please add more discussion and experiment details (electrolyte and current density) in this section.
- (4) The interfacial resistance of the Li/PEO-DLSL reduced by 2/3 compared with the traditional Li/PEO,

indicating a contribution of the hierarchical 3D anode structure, which is beneficial for the high current density charge/discharge performance. It will be very helpful if the author could provide the lithium plating/stripping data at a little higher current density, such as 2 mA/cm². It will be very promising if this 3D anode can provide stable lithium plating/stripping at 2 mA/cm², since the currently commercial battery generally own a capacity area density of 2mAh/cm².

(5) During the C-rate test of all-solid-state LiFePO₄ cells, it will be more convenient for the readers if the author could provide the current densities instead of the C-rate, such as the current density of 0.8mA/cm² corresponding to a 2C.

(6) It will be very helpful for investigating the stability of electrolyte and interface, if the impedance change could be monitored in the Li/Li symmetric cells or Li/LiFePO₄ solid-state cells.

Reviewer #3 (Remarks to the Author):

Review #NCOMM-18-35405-T

The presented manuscript gives detail investigations of using nature diatomite as template for hosting metallic Li for all solid state Li battery application. The fabricated hierarchical structured PEO-DLSL composite negative electrode shows excellent ability on suppressing Li dendrite formation and good battery performance. The results are well presented and of interest for battery researches. However, the Li diffusion coefficient in Si is much lower than Li self-diffusion which makes the major concept of the manuscript arguable. Nevertheless, some concerns of the manuscript that should be answered before it can be accepted for publication.

- On page 3 (PDF file page 3), line 3, authors refer to the dendrite formation is due to the high reactivity between Li metal and liquid electrolyte is not true.
- On page 4, second line from bottom. The third point as that shown the concept in figure 1b is arguable. The Li diffusion coefficient in Si is only range from 10⁻¹¹ to 10⁻¹³ cm²/s (Arie, DOI: 10.4028/www.scientific.net/DDF.326-328.87; Wen, J. solid state chem, 37 (1981) 271-278; Ding, solid state ionics, 180 (2009)222-225) which is similar or lower than Li self-diffusion 5x10⁻¹¹ cm²/s at RT. If a Li foil does not work, as shown in the manuscript, how could a system with slower Li diffusion coefficient works better? Do authors consider the major contribution is actually coming from the bigger contact surface area (especially lower resistance in EIS result) of DLSL particles and much better Li-ion conductive paths due to the embedded SPE in composite DLSL?
- On page 8, line 6. Instead of "based on the total electrode weight", life would be better if authors directly give the weight!
- Both DF-Si-Li0.5 and DF-Si-Li0.8 shows delithiation from Li_{4.4}Si but not DF-Si-Li0.2. Why?
- Page 15, line 3. A PEO-DLSL electrode with 240 micrometer in thickness to give a 4 mA h cm⁻² capacity does not show any benefit than using graphite!
- For the symmetric cell cycling, figure 5a for the current densities higher than 200 microA cm⁻², the voltages are slowly decrease to zero when the polarizations are switched. Why?
- For the Li stripping/plating process, could authors explain why Li will grow back into the diatomite templates? Naturally, the system would find the lowest resistance paths for both electrons and Li-ions to deposit Li atoms which are most likely next to the interface between solid electrolyte and negative electrode. With a soft SPE, why Li has to deposit along the hierarchical structure? What makes Li would like to deposit at some place far away from the interface, especially the Li diffusion path is much longer?

The point-to-point responses to the reviewers' questions

We have carefully considered the valuable comments and suggestions from all the reviewers, and have addressed them here and made revisions in the manuscript accordingly. For clearness, responses have been marked with RED color and started with “**”.

Reviewer 1

This work introduces a new concept to design Li-metal based composite anode. The performance is good, and this work can be applied to current battery fabrication process. Here I have some questions to help authors further improve the quality of their work. My comments are given below:

**** We thank the reviewer's valuable comments for improving the quality of our manuscript.**

1. The motivation of $\text{Li}_{4.4}\text{Si}$ framework design is not clear. Please add more detailed discussion in the manuscript.

**** Thank the reviewer for very nice suggestion here. We consider that the $\text{Li}_{4.4}\text{Si}$ framework in our designed solid composite anode exhibits benefits in the following several aspects. First, the porous $\text{Li}_{4.4}\text{Si}$ framework could act as the host to accommodate Li in the hierarchical pores, resulting in robust electrode structure for electrochemical plating/stripping of Li. Second, the three-dimensional (3D) $\text{Li}_{4.4}\text{Si}$ framework with large electrode-solid electrolyte interfacial surface area could reduce the localized current density and suppress the dendrite growth of Li. Third, with the help of PEO-SPE decoration in the 3D framework, Li could deposit back into the rigid $\text{Li}_{4.4}\text{Si}$ framework, thus maintaining the electrode stability. Finally, the abundant and sustainable diatomite provide low cost source to fabricate $\text{Li}_{4.4}\text{Si}$ framework via facile magnesiothermic reduction conversion from SiO_2 to Si. Taking aforementioned merits into consideration, we believe that the $\text{Li}_{4.4}\text{Si}$ framework derived from diatomite is unique to construct the advanced Li composite anode. *We added detailed discussion of the function of $\text{Li}_{4.4}\text{Si}$ framework in the manuscript (discussion part, page 20, line 11-22 and page 21, line 1-5).***

2. “In this composite anode, the elastic PEO-SPE could enhance the contact area between the hybrid anode and solid electrolyte, shortening ion transportation pathway and homogenizing the ion flux in the solid composite electrode.” please specify the motivation of adding PEO-based electrolyte in the anode. What's the ionic conductivity for the composite anode and the lithium silicide-Li hybrid material?

**** We thank the reviewer's constructive comments/suggestions here. The poor interfacial contact between the electrode and solid electrolyte is a long term issue for all-solid-state Li metal batteries. Limited contact area would result in large interfacial**

resistance and high voltage polarization. The elastic and soft PEO-SPE with good Li^+ ion conductivity (4.63×10^{-4} S/cm at 60 °C, Supplementary Figure 10) could construct a continuous ion transport channel throughout the entire electrode, thus homogenize the ion flux in the solid composite anode enabling the Li plating/stripping inside the framework. It is hard to directly measure the ionic conductivity of electrodes. The “ionic conductivity of the composite anode” in our work was indicated to the Li^+ ion diffusion and transportation capability in the PEO-SPE/Li-Si composite framework. According to the electrochemical impedance spectroscopy (EIS) results of symmetric and full cells (Figure R1), PEO-DLSL composite electrode exhibited smaller charge-transfer resistance (R_{ct}) before and after cycling than Li foil, further confirming the advantage of our design. *We have included these results in the revised manuscript and supplementary materials (page 20, line 11-22 and page 21, line 1-5 in the manuscript; page 18, Supplementary Figure 15).*

Figure R1. a) EIS curves of Li/LiFePO₄ and PEO-DLSL/LiFePO₄ full cells before and after 10 cycles. b) EIS curves of Li/Li and PEO-DLSL/PEO-DLSL symmetric cells before and after 20 cycles.

3. Please explain why the hybrid anode is more stable than the pure Li metal anode in symmetric cell cycling. And authors compared the interfacial impedance in both anodes, please specify what types of interfacial impedance are indicated in the manuscript.

** We thank the reviewer’s critical comments and nice suggestions here. In our opinion, the enhanced symmetric cell performance could be owing to three parts. First, Li was embedded in the hierarchical pores, leading to small local current density and uniform Li^+ ion flux, which could suppress the dendritic growth and ‘dead’ Li formation. Second, the PEO-SPE decorated on Li_xSi framework exhibited large interfacial contact area and high Li^+ ion conductivity to enable the Li stripping/plating inside the framework rather than the surface of electrode. Third, the 3D porous Li_xSi /PEO-SPE framework with good mechanical strength can support the electrode with negligible volume change during cycling, which ensured intimate interfacial

contact to further maintain stable cycling performance. In the Nyquist plots, the impedance spectra of the cells are comprised of two semicircles and one inclined line, corresponding to the electrolyte resistance (R_e), charge transfer resistance (R_{ct}) and the Warburg impedance, respectively. In our manuscript, the ‘interfacial impedance’ was indicated to the R_{ct} . As shown in the Figure R1, the PEO-DLSL electrode demonstrated smaller R_{ct} in both symmetric and full cells, indicating smaller interfacial impedance. *We have included above discussion in the revised manuscript (discussion part, Page 20, line 8-22 and page 21, line 1-5).*

4. In the all solid state battery, there is no electrolyte in the cathode, how to maintain proper ionic conductivity in cathode part in order to facilitate a full cell cycling?

**** We thank the reviewer for critical comments here. For preparing the working cathode, PEO-SPE was firstly made by mixing PEO and LiTFSI in anhydrous acetonitrile (Energy Chemical, www.energy-chemical.com) and used as binder. Then, LiFePO₄ (MTI Inc) powder, super-P acetylene black (Alfa Aesar) and PEO-SPE were mixed at a weight ratio of 80:10:10 in the N-Methyl-2-pyrrolidone (NMP, Sigma-Aldrich) solvent to form a uniform slurry. Afterwards, the slurry was casted on an Al-carbon foil (MTI) by a doctor blade and dried at 60 °C in vacuum. Therefore, the surface of the LiFePO₄ particles was uniformly decorated by the high Li⁺ conductivity PEO-SPE in our fabricated cathode. Besides, the plastic PEO-SPE could maintain intimate interfacial contact between cathode and the solid electrolyte layer. The aforementioned merits together guaranteed the rapid Li⁺ transportation in the cathode part. *We included the details in the Method part (Page 25, line 15-19) in the revised manuscript.***

Reviewer 2

In this paper, the hierarchical silicon containing 3D anode was prepared and investigated in PEO based all-solid-state batteries. With this composite Si-Li alloy electrode, the anode/polymer-electrolyte interface was increased and therefore the area current density was lower, which are beneficial for homogenization of the Li⁺ flux across the interface and suppression of lithium dendrite at larger current density plating, two biggest challenges of the solid-state batteries. Higher current density performance verified the advantage of this specific anode design. In my opinion, this paper can be published after addressing following issues.

**** We thank the reviewer’s positive comments to our work.**

1. It has been reported that the flat lithium metal anode could also give stable long-term plating/stripping at lower current density, such as lower than 0.5 mA/cm². The advantage of this 3D anode in solid-state cells will become significant in higher current density. A description of “higher current density plating/stripping” should be added in Figure 1b and 1c.

**** We appreciated the reviewer for critical comments/suggestions for helping us**

improve this manuscript. As shown in Figure R2b and R2c, we have added the suggested description of “high current density ($>0.5 \text{ mA/cm}^2$) plating/stripping”. We have included above correction in the revised manuscript (Page 6, Figure 1).

Figure R2. Schematic illustration of the PEO-DLSL anode fabrication process and Li stripping/plating behaviors in comparison to planar Li anode. a, The fabrication process of PEO-DLSL. b-c, Li stripping/plating behavior in all solid state Li metal batteries with PEO-DLSL anode and planar Li foil anode, respectively.

2. “the extracted specific capacity of DF-SiO-Li and DF-Li electrode was $\sim 1031 \text{ mAh/g}$ and $\sim 924 \text{ mAh/g}$, with only $\sim 399 \text{ mAh/g}$ and $\sim 265 \text{ mAh/g}$ of capacity contributed from the Li, respectively” It is hard to understand that only 265 mAh/g of capacity contributed from the Li in the DF-Li, since the DF itself should not contribute obvious capacity. Please revise the description here and reduce the misunderstanding of the readers.

** We thank the reviewer’s very careful comments here. In our work, the ‘capacity contributed from the Li metal’ was defined as the capacity exhibited in the charge voltage plateau that below the 0.2 V . In the composite electrode fabrication process, a part of Li was consumed to react with DF, DF-SiO, or DF-Si in the first place. The lithiation product was $\text{Li}_{4.4}\text{Si-Li}_2\text{O}$ composite for DF and DF-SiO, and $\text{Li}_{4.4}\text{Si}$ for DF-Si, respectively. After the full lithiation, the extra Li was localized and embedded into the hierarchical pores of $\text{Li}_{4.4}\text{Si-Li}_2\text{O}$ or $\text{Li}_{4.4}\text{Si}$ framework. In the product of lithiated DF, most of Li was consumed by the DF framework with least amount of Li left in the framework and thus contributed to the smallest capacity (265 mAh/g). During the charge process to evaluate the electrode specific capacity, Li metal was stripped from the composite anode in the first place due to its lowest potential. The de-lithiation of $\text{Li}_{4.4}\text{Si}$ started when the voltage reached $\sim 0.2 \text{ V}$. To reduce the

misunderstanding, we revised the 'Li' as the 'Li metal' and have added a black dash line in Figure R3 to visually demonstrate the capacity contributed from Li metal (Page 9, line 3; Page 13, line 12 in the manuscript and Page 5, Supplementary Figure 2).

Figure R3. Li stripping performance of lithiated diatomite, DF-SiO and DF-Si in liquid electrolyte cells.

3. In Figure 3h, the DF-Si-Li alloys showed relatively low de-lithiation capacity compared with the previously reported Li_xSi alloys, such as the DF-Si- $\text{Li}_{0.2}$ didn't give visible charge capacity when charge to 1.0 V vs Li/Li^+ , the DF-Si- $\text{Li}_{0.5}$ give a capacity of 332 mAh/g with a voltage plateau at 0.4-0.5 V without de-lithiation capacity between 0.1-0.4 V. These features might because the lithium consumption in the formation of SEI in traditional liquid electrolyte. Please add more discussion and experiment details (electrolyte and current density) in this section.

****** We thank the reviewer's very nice comments/suggestions here. To achieve full lithiation, 0.5 g of silicon needs about 0.5 g of Li according to the chemical stoichiometry. The lithiated products of DF-Si- $\text{Li}_{0.2}$ and DF-Si- $\text{Li}_{0.5}$ were Li_xSi and $\text{Li}_{4.4}\text{Si}$, respectively. The electronic conductivity of Li_xSi was low ($3.06 \times 10^4 \Omega \cdot \text{cm}$, W. Tang et al., *Adv. Mater.* 2018, 1801745) and the fabricated electrode was thick (~240 μm). Therefore, the electron transfer in DF-Si- $\text{Li}_{0.2}$ electrode is heavily limited, leading to low extracted charge capacity of DF-Si- $\text{Li}_{0.2}$. For DF-Si- $\text{Li}_{0.5}$ electrode, the initially embedded metallic Li could conduct electron, resulting in better electrochemical performance. When Li continuously extracted from the DF-Si- $\text{Li}_{0.5}$ electrode, the electron conductivity of DF-Si- $\text{Li}_{0.5}$ electrode kept on getting worse, which could be revealed by the curve fluctuation in Figure 3h. In addition, the consumption of Li in the formation of SEI in the liquid electrolyte may further reduce the charge capacity of DF-Si-Li alloys. The electrolyte used for Li extraction was 1M

LiPF₆ in EC/DEC (3:7, v:v) and the current density was 10 $\mu\text{A}/\text{mg}$. We have included this discussion and experimental details in the revised manuscript (Page 22, line 3-11; Page 24, line 19-22).

4. The interfacial resistance of the Li/PEO-DLSL reduced by 2/3 compared with the traditional Li/PEO, indicating a contribution of the hierarchical 3D anode structure, which is beneficial for the high current density charge/discharge performance. It will be very helpful if the author could provide the lithium plating/stripping data at a little higher current density, such as 2 mA/cm^2 . It will be very promising if this 3D anode can provide stable lithium plating/stripping at 2 mA/cm^2 , since the currently commercial battery generally own a capacity area density of 2 mAh/cm^2 .

** We thank the reviewer's nice suggestion here. We tested the Li plating/stripping behavior at 2 mA/cm^2 . As shown in Figure R4, symmetric cells based on Li foil and PEO-DLSL electrodes were cycled at 0.2 mA/cm^2 to activate the cells at first. Subsequently, the current density was increased to 2 mA/cm^2 . The polarization voltage of Li foil rapidly exceeded the safety voltage limit (5V). In contrast, PEO-DLSL demonstrated stable cycling performance over 95 hours with polarization voltage around 4 V. It is worth noting that although the PEO-DLSL can be operated at 2 mA/cm^2 the overpotential is a little high, which needs further optimization in future. We have included the above result in the revised manuscript and supplementary material (Page 17, line 7-9 in the manuscript; Supplementary Figure 13).

Figure R4. Polarization voltage of Li and PEO-DLSL symmetric cells cycled at a current density of 2 mA/cm^2 .

5. During the C-rate test of all-solid-state LiFePO₄ cells, it will be more convenient for the readers if the author could provide the current densities instead of the C-rate, such as the current density of 0.8 mA/cm^2 corresponding to a 2C.

** We thank the reviewer's good comment. As shown in Figure R5e, we have added the numerical value of current density (1C = 170 mA/g , corresponding to 0.4 mA/cm^2)

in the revised manuscript and Figure 5e. The corresponding modification could be found in the revised manuscript (Page 18, Figure 5; Page 19, line 9).

Figure R5. Electrochemical performance comparison of the cells using PEO-DLSL and planar Li foil anode, respectively.

6. It will be very helpful for investigating the stability of electrolyte and interface, if the impedance change could be monitored in the Li/Li symmetric cells or Li/LiFePO₄ solid-state cells.

** We thank the reviewer's great suggestion here. We investigated the interfacial impedance variation of symmetric and full cells using PEO-DLSL and Li foil, and the results are shown in Figure R6. In the Nyquist plots, the electrochemical impedance spectra (EIS) of the cells are comprised of two semicircles and one inclined line, corresponding to the electrolyte resistance (R_e), charge transfer resistance (R_{ct}) and the Warburg impedance, respectively. Here, the interfacial impedance was indicated to the R_{ct} . According to the EIS results, the original areal R_{ct} of PEO-DLSL/LiFePO₄ was $\sim 116 \Omega \text{ cm}^{-2}$, and it changed to $\sim 126 \Omega \text{ cm}^{-2}$ after 10 cycles for full cells. Significantly, the areal R_{ct} of Li/LiFePO₄ changed from $\sim 153 \Omega \text{ cm}^{-2}$ to $\sim 212 \Omega \text{ cm}^{-2}$, indicating more stable interface in PEO-DLSL based cell. For symmetric cells, the original areal R_{ct} of PEO-DLSL/PEO-DLSL and Li/Li was $\sim 124 \Omega \text{ cm}^{-2}$ and $\sim 292 \Omega \text{ cm}^{-2}$, respectively. After 20 cycles, the corresponding areal R_{ct} was $\sim 138 \Omega \text{ cm}^{-2}$ and $\sim 307 \Omega \text{ cm}^{-2}$, respectively. The above results clearly demonstrate a more stable electrode/electrolyte interface and smaller interfacial resistance induced by the PEO-DLSL electrode design. We have included these results and discussion in the revised manuscript and supporting information (Page 20, line 16-22 in the manuscript; Page 18, Supplementary Figure 15 in the supporting information).

Figure R6. a) EIS curves of Li/LiFePO₄ and PEO-DLSL/LiFePO₄ full cells before and after 10 cycles. b) EIS curves of Li/Li and PEO-DLSL/PEO-DLSL symmetric cells before and after 20 cycles.

Reviewer 3

The presented manuscript gives detail investigations of using nature diatomite as template for hosting metallic Li for all solid stat Li battery application. The fabricated hierarchical structured PEO-DLSL composite negative electrode shows excellent ability on suppressing Li dendrite formation and good battery performance. The results are well presented and of interest for battery researches. However, the Li diffusion coefficient in Si is much lower than Li self-diffusion which makes the major concept of the manuscript arguable. Nevertheless, some concerns of the manuscript

that should be answered before it can be accepted for publication.

****We thank the reviewer's suggestive comments for improving the quality of our manuscript.**

1. On Page 3 (PDF file Page 3), line 3, authors refer to the dendrite formation is due to the high reactivity between Li metal and liquid electrolyte is not true.

**** We thank the reviewer's very careful comments here. As the reviewer pointed out that the dendrite formation cannot be directly attributed to the high reactivity between Li metal and liquid electrolyte. As we know, Li plating is conducted based on the chemical equation below.**

For liquid organic electrolytes, the Li deposition includes at least two steps: liquid phase mass transfer step (LPMTS) and electron exchange step (EES). Li^+ was transferred from liquid electrolytes to the surface of electrode through LPMTS, while electron was conducted to the surface through current collector and Li. The Li^+ conductivity of electrolytes is order of magnitude lower than the electrical conductivity of current collector and Li. Therefore, LPMTS is the rate determine step during Li deposition. In practical cells, LPMTS is carried out by a convective diffusion process. At different zones, the mass transfer rate and Li^+ flux have discrepancy. Dendrite appears at where mass transfer rate is high due to different Li deposition rate. Once the dendrite growth starts, the gap between mass transfer aggravate, resulting in more serious inhomogeneous deposition. Hence, the dendrite growth of Li is highly related to the homogeneity of Li^+ flux at the interface of Li metal anode and electrolyte. The high reactivity between Li metal and liquid electrolyte makes the uncontrollable formation of SEI resulting in obvious fluctuation of Li^+ flux at the surface of Li metal anode to accelerate the growth of Li dendrites. *In the revised manuscript, we changed the description of 'The high reactivity between Li metal anode and liquid electrolyte raises up the growth of dendrites' to 'The heterogeneous and fragile solid electrolyte interphase (SEI) caused by high reactivity between Li metal anode and liquid electrolyte raises up the growth of dendrite'.*

2. On Page 4, second line from bottom. The third point as that shown the concept in figure 1b is arguable. The Li diffusion coefficient in Si is only range from 10^{-11} to 10^{-13} cm^2/s (Arie, DOI: 10.4028/www.scientific.net/DDF.326-328.87; Wen, J. solid state chem, 37 (1981) 271-278; Ding, solid state ionics, 180 (2009)222-225) which is similar or lower than Li self-diffusion 5×10^{-11} cm^2/s at RT. If a Li foil does not work, as shown in the manuscript, how could a system with slower Li diffusion coefficient works better? Do authors consider the major contribution is actually coming from the bigger contact surface area (especially lower resistance in EIS result) of DLSL particles and much better Li-ion conductive paths due to the embedded SPE in composite DLSL?

**** We thank the reviewer's nice comment here. The hierarchical 3D composite anode design is essential to its better performance in comparison to Li metal foil anode. As the reviewer pointed out that the bigger contact surface area in hierarchical 3D**

composite anode to decrease local current density and homogenize Li^+ ion flux in the electrode. To enable high efficiency of the hierarchical 3D electrode, the high Li^+ ion conductivity inside the electrode is necessary. In our fabricated composite anode, the Li^+ ion conductivity is both contributed by the PEO-SPE decoration layer and the $\text{Li}_{4.4}\text{Si}$ framework. It is well known that PEO-SPE is a good Li^+ ion conductor when worked at 60 °C showing high Li^+ ion conductivity of 4.63×10^{-4} S/cm at 60 °C (revealed in Supplementary Figure 10 in the revised supporting information). This means that the Li^+ ions can be conducted along the surface of $\text{Li}_{4.4}\text{Si}$ framework in as-fabricated hierarchical 3D composite anode. The key thing is that whether the $\text{Li}_{4.4}\text{Si}$ framework can conduct the Li^+ ions. We investigated the references provided by the reviewer, in which the Li^+ ion diffusion coefficient in Si was measured based on the polycrystalline silicon film. The results in these literatures are not consistent with our systems that the frameworks are composed of amorphous $\text{Li}_{4.4}\text{Si}$. In addition, we found that some literatures (Z.G. Zhang et al., *ChemElectroChem*, 2015, **2**, 1292-1297; S.C. Jung et al., *J. Phys. Chem. Lett.* 2014, **5**, 1283–1288; C.Y. Chou et al., *Surf. Sci.* 2013, **612**, 16-23) reported diffusion coefficients of Li in Li–Si alloys in the range between 10^{-9} cm²/S and 10^{-7} cm²/S at room temperature, which were all higher than Li self-diffusion 5×10^{-11} cm²/s at room temperature. These inconsistent results make it hard to judge the Li^+ ion conduction in the $\text{Li}_{4.4}\text{Si}$ framework. In this case, we adopted the EIS comparison of $\text{Li}_{4.4}\text{Si}$ electrode and PEO-SPE decorated $\text{Li}_{4.4}\text{Si}$ (PEO- $\text{Li}_{4.4}\text{Si}$) electrode to reveal the Li^+ ion conduction role of $\text{Li}_{4.4}\text{Si}$ framework. We fabricated $\text{Li}_{4.4}\text{Si}$ and PEO- $\text{Li}_{4.4}\text{Si}$ pallets (diameter: 12 mm, thickness: ~240 μm) by cold-pressing. These two pallets were sandwiched by a thin layer of PEO electrolyte and stainless steel electrode to measure their resistance via electrochemical impedance spectroscopy. As shown in Figure R7, the areal resistance of $\text{Li}_{4.4}\text{Si}$ electrode was measured as ~54 Ω cm², which means that the Li^+ ions can be conducted in the $\text{Li}_{4.4}\text{Si}$ framework. After making decoration of PEO on the framework, the measured areal resistance reduced a little to ~37 Ω cm², indicating that the PEO decoration could enhance the Li^+ ion conductivity of PEO- $\text{Li}_{4.4}\text{Si}$ electrode. On the basis of above results, we consider that the Li^+ ion conduction in as-fabricated hierarchical 3D composite anode is endowed by both PEO-SPE decoration layer and $\text{Li}_{4.4}\text{Si}$ framework. *We have included the above discussion and data in the revised manuscript and supporting information (discussion part, page 21, line 6-22 in the manuscript; Page 19, Supplementary Figure 16 in the supporting information).*

Figure R7. Nyquist plots of $\text{Li}_{4.4}\text{Si}$ and $\text{PEO-Li}_{4.4}\text{Si}$ at 60 °C. Blue dots is corresponding to $\text{Li}_{4.4}\text{Si}$, while red circle is $\text{PEO-Li}_{4.4}\text{Si}$.

3. On Page 8, line 6. Instead of “based on the total electrode weight”, life would be better if authors directly give the weight!

** We thank the reviewer’s good suggestion. We have added the value of the electrode weight in the revised manuscript. *The corresponding modification could be found in the revised manuscript (Page 24, line 17).*

4. Both DF-Si- $\text{Li}_{0.5}$ and DF-Si- $\text{Li}_{0.8}$ shows delithiation from $\text{Li}_{4.4}\text{Si}$ but not DF-Si- $\text{Li}_{0.2}$. Why?

** We thank the reviewer’s very important question to us. In our fabrication process, Li was firstly consumed for the lithiation of silicon to construct Li_xSi framework. According to the chemical stoichiometry, there was no metallic Li left in the DF-Si- $\text{Li}_{0.2}$ electrode. Due to the poor electronic conductivity of Li_xSi ($\sim 3.06 \times 10^4 \Omega \cdot \text{cm}$, W. Tang et al., *Adv. Mater.*, 2018, 1801745) and thickness of the electrodes ($\sim 240 \mu\text{m}$), the electron transfer in DF-Si- $\text{Li}_{0.2}$ electrode is heavily limited, leading to lowest extracted charge capacity of DF-Si- $\text{Li}_{0.2}$. For DF-Si- $\text{Li}_{0.5}$ and DF-Si- $\text{Li}_{0.8}$ electrodes, the initially embedded metallic Li could conduct electrons, resulting in better electrochemical performance. With Li continuously extract from the DF-Si- $\text{Li}_{0.5}$ electrode, the electron conductivity of DF-Si- $\text{Li}_{0.5}$ electrode kept on getting worse, which could be revealed by the curve fluctuation in Figure 3h in the manuscript. In addition, the consumption of Li in the formation of SEI in the liquid electrolyte may further reduce the charge capacity of DF-Si-Li alloys. *We have included the explanation in the discussion part in the revised manuscript and supporting information (Page 22, line 1-14; Page 20, Supplementary Figure 17 in supporting information).*

5. Page 15, line 3. A PEO-DLSL electrode with 240 micro-meter in thickness to give a 4 mAh cm^{-2} capacity does not show any benefit than using graphite!

** We thank the reviewer’s nice comment here. In present work, we stripped/plated the Li of 4 mAh cm^{-2} to show the good stripping/plating behavior in the hierarchical 3D electrode without the inhomogeneous deposition of Li as revealed on the surface

of Li metal foil anode. The specific capacity of as-fabricated composite anode has been tested as 1153 mAh/g. To make this composite anode comparable to commercial graphite anode, it needs to decrease the thickness down to 100 μm . Due to the limited instruments in the laboratory, 240 μm is a moderate thickness for our fabricated PEO-DLSL electrode. If the thickness is below 100 μm , the electrode was hard to be peeled off from the tableting die. However, with using advanced rolling machine for Li composite anode manufacture, it can be anticipated to fabricate thinner PEO-DLSL electrode to make it more beneficial than graphite anode.

6. For the symmetric cell cycling, figure 5a for the current densities higher than 200 $\mu\text{A cm}^{-2}$, the voltages are slowly decrease to zero when the polarizations are switched. Why?

*** We thank the reviewer's nice comment here. This phenomena in the voltage curve is caused by the program we set for the cell cycling. It can be seen that the polarization voltage of Li metal foil electrode increased rapidly when the current density increased up to 200 $\mu\text{A cm}^{-2}$. To release this polarization, we set the standing time as thirty minutes for the symmetric cell cycling when the current densities were higher than 200 $\mu\text{A cm}^{-2}$. As a result, the voltage slowly decreased to zero at the end of charge/discharge process. We have included the explanation in the in the revised manuscript (Page 25, line 9-11).*

7. For the Li stripping/plating process, could authors explain why Li will grow back into the diatomite templates? Naturally, the system would find the lowest resistance paths for both electrons and Li-ions to deposit Li atoms which are most likely next to the interface between solid electrolyte and negative electrode. With a soft SPE, why Li has to deposit along the hierarchical structure? What makes Li would like to deposit at some place far away from the interface, especially the Li diffusion path is much longer?

*** We thank the reviewer's very nice comment here. As the reviewer pointed out that Li atoms would be deposited at where the sufficient Li^+ ions and electrons are both provided. Usually, in the traditional electrode, the electronic conductivity is much higher than that of Li^+ ions and thus the deposition of Li is likely to be appeared at the interface between solid electrolyte and the anode. But in our fabricated composite electrode, the Li^+ ion conductivity is dominant to make the deposition of Li occurring in the electrode framework. As shown in Figure R8, the Li^+ ions can be conducted in the PEO-SPE coated $\text{Li}_{4.4}\text{Si}$ -PEO framework with high conductivity but the electrons are hard to be conducted in the framework due to its low electronic conductivity. The electron transfer is likely to be conducted in the metallic Li. Therefore, Li would prefer to deposit back into the $\text{Li}_{4.4}\text{Si}$ -PEO framework (denoted in red dash circle) along the hierarchical structure rather than the electrolyte/electrode interface in our fabricated composite anode. We have included the explanation in the discussion part in the revised manuscript and supporting information (Page 22, line 1-13; Page 20, Supplementary Figure 17 in supporting information).*

Figure R8. Schematic illustration of Li deposition mechanism in our fabricated hierarchical composite anode.

Reviewers' comments:

Reviewer #1 (Remarks to the Author):

The authors have addressed my concerns and questions, I would like to suggest "Acceptance" for this revised manuscript.

Reviewer #2 (Remarks to the Author):

The authors have revised the manuscript according to three reviewers' comments and suggestions in this revision. In this reviewer's opinion, this revised version can be accepted.

Reviewer #3 (Remarks to the Author):

please see the attached file for the comments.

Thank the authors for the detail answers of my questions about this manuscript. The response from the authors had been well received and understood. For those questions that reviewer is satisfied were removed from the old review text while those questions are not really answered by the authors are keep below for further discussion before it can be accepted for publication. My new comments are typed in color blue.

Reviewer 3

The presented manuscript gives detail investigations of using nature diatomite as template for hosting metallic Li for all solid stat Li battery application. The fabricated hierarchical structured PEO-DLSL composite negative electrode shows excellent ability on suppressing Li dendrite formation and good battery performance. The results are well presented and of interest for battery researches. However, the Li diffusion coefficient in Si is much lower than Li self-diffusion which makes the major concept of the manuscript arguable. Nevertheless, some concerns of the manuscript that should be answered before it can be accepted for publication.

5. Page 15, line 3. A PEO-DLSL electrode with 240 micro-meter in thickness to give a 4 mAh cm^{-2} capacity does not show any benefit than using graphite!

** We thank the reviewer's nice comment here. In present work, we stripped/plated the Li of 4 mAh cm^{-2} to show the good stripping/plating behavior in the hierarchical 3D electrode without the inhomogeneous deposition of Li as revealed on the surface of Li metal foil anode. The specific capacity of as-fabricated composite anode has been tested as 1153 mAh/g . To make this composite anode comparable to commercial graphite anode, it needs to decrease the thickness down to $100 \mu\text{m}$. Due to the limited instruments in the laboratory, $240 \mu\text{m}$ is a moderate thickness for our fabricated PEO-DLSL electrode. If the thickness is below $100 \mu\text{m}$, the electrode was hard to be peeled off from the tableting die. However, with using advanced rolling machine for Li composite anode manufacture, it can be anticipated to fabricate thinner PEO-DLSL electrode to make it more beneficial than graphite anode.

What I mean is that graphite anode usually only needs a thickness of < 100 micrometer to give an energy density of 4 mA h cm^{-2} in a commercial LIB while it needs 240 micrometer for the PEO-DLSL electrode. Even though authors can reduce the thickness down to 100 micrometer, the energy density would decrease accordingly. Since the specific capacity of the composite anode is much higher than graphite, we can assume this is only a technical problem and can be ignored.

6. For the symmetric cell cycling, figure 5a for the current densities higher than $200 \text{ microA cm}^{-2}$, the voltages are slowly decrease to zero when the polarizations are switched. Why?

** We thank the reviewer's nice comment here. This phenomena in the voltage curve is caused by the program we set for the cell cycling. It can be seen that the polarization voltage of Li metal foil electrode increased rapidly when the current

density increased up to $200 \mu\text{A cm}^{-2}$. To release this polarization, we set the standing time as thirty minutes for the symmetric cell cycling when the current densities were higher than $200 \mu\text{A cm}^{-2}$. As a result, the voltage slowly decreased to zero at the end of charge/discharge process. *We have included the explanation in the in the revised manuscript (Page 25, line 9-11).*

Here authors actually did not answer the question! The slowly decreasing of voltage is more obvious for PEO-DLSL than Li foil in figure 5a. The rapidly increasing of voltage simply means the polarization (resistance) of the cell is increasing rapidly. When the system was set to “standing” mode, a 0 V should be measured immediately instead of slowly decreasing of voltage ($V=IR$, when $I=0$, V has to be 0) for a symmetric cell. If the concentration of Li is different on two electrodes due to the stripping/deposition process to give a voltage different, then a slowly decreasing of voltage means a leakage through the electrolyte. Authors should give a better comment on this question.

7. For the Li stripping/plating process, could authors explain why Li will grow back into the diatomite templates? Naturally, the system would find the lowest resistance paths for both electrons and Li-ions to deposit Li atoms which are most likely next to the interface between solid electrolyte and negative electrode. With a soft SPE, why Li has to deposit along the hierarchical structure? What makes Li would like to deposit at some place far away from the interface, especially the Li diffusion path is much longer?

****** We thank the reviewer’s very nice comment here. As the reviewer pointed out that Li atoms would be deposited at where the sufficient Li^+ ions and electrons are both provided. Usually, in the traditional electrode, the electronic conductivity is much higher than that of Li^+ ions and thus the deposition of Li is likely to be appeared at the interface between solid electrolyte and the anode. But in our fabricated composite electrode, the Li^+ ion conductivity is dominant to make the deposition of Li occurring in the electrode framework. As shown in Figure R8, the Li^+ ions can be conducted in the PEO-SPE coated $\text{Li}_{4.4}\text{Si}$ -PEO framework with high conductivity but the electrons are hard to be conducted in the framework due to its low electronic conductivity. The electron transfer is likely to be conducted in the metallic Li. Therefore, Li would prefer to deposit back into the $\text{Li}_{4.4}\text{Si}$ -PEO framework (denoted in red dash circle) along the hierarchical structure rather than the electrolyte/electrode interface in our fabricated composite anode. *We have included the explanation in the discussion part in the revised manuscript and supporting information (Page 22, line 1-13; Page 20, Supplementary Figure 17 in supporting information).*

Figure R8. Schematic illustration of Li deposition mechanism in our fabricated hierarchical composite anode.

The answer seems reasonable but rather intuitively. I agree that Li will deposit firstly on the two-phase boundary (the red dash circle). However, Li is a solid metal instead of liquid which has a very low Li diffusion of $5 \times 10^{-11} \text{ cm}^2/\text{s}$. This means Li will only deposit as a very thin film on the surface of hierarchical composite and grow long the hierarchical composite to the interface, instead of fill up the gaps between the hierarchical composite. Once the Li on the surface of hierarchical composite grows to the interface of electrode/electrolyte, Li will only deposit (accumulate) at the interface because the shortest distance for Li ions to travel. Again, Li is a solid metal instead of liquid. Why it has to fill up the gap faster than grow long the composite with such a low self diffusion?

The point-to-point responses to the reviewers' questions

We have carefully considered the valuable comments and suggestions from the reviewer 3, and have addressed them here and made revisions in the manuscript accordingly. For clearness, our newest responses have been marked with purple color and started with “**”.

Thank the authors for the detail answers of my questions about this manuscript. The response from the authors had been well received and understood. For those questions that reviewer is satisfied were removed from the old review text while those questions are not really answered by the authors are keep below for further discussion before it can be accepted for publication. My new comments are typed in color blue.

Reviewer 3

The presented manuscript gives detail investigations of using nature diatomite as template for hosting metallic Li for all solid stat Li battery application. The fabricated hierarchical structured PEO-DLSL composite negative electrode shows excellent ability on suppressing Li dendrite formation and good battery performance. The results are well presented and of interest for battery researches. However, the Li diffusion coefficient in Si is much lower than Li self-diffusion which makes the major concept of the manuscript arguable. Nevertheless, some concerns of the manuscript that should be answered before it can be accepted for publication.

5. Page 15, line 3. A PEO-DLSL electrode with 240 micro-meter in thickness to give a 4 mAh cm⁻² capacity does not show any benefit than using graphite!

**** We thank the reviewer's nice comment here. In present work, we stripped/plated the Li of 4 mAh cm⁻² to show the good stripping/plating behavior in the hierarchical 3D electrode without the inhomogeneous deposition of Li as revealed on the surface of Li metal foil anode. The specific capacity of as-fabricated composite anode has been tested as 1153 mAh/g. To make this composite anode comparable to commercial graphite anode, it needs to decrease the thickness down to 100 μm. Due to the limited instruments in the laboratory, 240 μm is a moderate thickness for our fabricated PEO-DLSL electrode. If the thickness is below 100 μm, the electrode was hard to be peeled off from the tableting die. However, with using advanced rolling machine for Li composite anode manufacture, it can be anticipated to fabricate thinner PEO-DLSL electrode to make it more beneficial than graphite anode.**

What I mean is that graphite anode usually only needs a thickness of < 100 micrometer to give an energy density of 4 mA h cm⁻² in a commercial LIB while it needs 240 micro-meter for the PEO-DLSL electrode. Even though authors can reduce the thickness down to 100 micro-meter, the energy density would decrease accordingly. Since the specific capacity of the composite anode is much higher than graphite, we can assume this is only a technical problem and can be ignored.

****We thank the reviewer's nice comment here. We further evaluate the areal capacity**

of our fabricated solid PEO-DLSL composite anode. According to the specific capacity contributed from Li (denoted as SC_{Li} , 838 mAh/g) and mass (~60 mg) of the round PEO-DLSL electrode disk (12 mm), the areal capacity (AC) could be calculated.

$$AC_{PEO-DLSL} = \frac{SC_{Li} \times Mass}{Area} = \frac{838 \text{ mAh/g} \times 0.060 \text{ g}}{3.14 \times 0.6^2 \text{ cm}^2} = 44.5 \text{ mAh/cm}^2$$

It's true that the capacity utilization ratio of the 240 μm PEO-DLSL anode is low when cycled with only 4 mAh cm^{-2} areal capacity. Take parameters affecting the areal capacity into account (such as specific capacity, areal mass loading, electrode thickness and porosity), 4 mAh cm^{-2} is a high areal capacity for the state-of-the-art commercial transition metal oxides cathode materials. To enhance the utilization efficiency of PEO-DLSL, higher areal capacity cathodes (Li-S, Li-O₂, Li-CO₂) and optimized thinner anodes could be effective strategies. Although novel cathodes always demonstrate very high theoretical specific capacity and energy density (2500 Wh/kg for Li-S batteries, 5200 Wh/kg for Li-O₂ batteries), their commercial application was still hindered by many fundamental issues. By contrast, thinner anodes would be more realizable and compatible for industrialization. Therefore, as the reviewer mentioned above, this is a technical problem which will be hopefully addressed in future.

6. For the symmetric cell cycling, figure 5a for the current densities higher than 200 microA cm^{-2} , the voltages are slowly decrease to zero when the polarizations are switched. Why?

**** We thank the reviewer's nice comment here. This phenomena in the voltage curve is caused by the program we set for the cell cycling. It can be seen that the polarization voltage of Li metal foil electrode increased rapidly when the current density increased up to 200 $\mu\text{A cm}^{-2}$. To release this polarization, we set the standing time as thirty minutes for the symmetric cell cycling when the current densities were higher than 200 $\mu\text{A cm}^{-2}$. As a result, the voltage slowly decreased to zero at the end of charge/discharge process. We have included the explanation in the in the revised manuscript (Page 25, line 9-11).**

Here authors actually did not answer the question! The slowly decreasing of voltage is more obvious for PEO-DLSL than Li foil in figure 5a. The rapidly increasing of voltage simply means the polarization (resistance) of the cell is increasing rapidly. When the system was set to "standing" mode, a 0 V should be measured immediately instead of slowly decreasing of voltage ($V=IR$, when $I=0$, V has to be 0) for a symmetric cell. If the concentration of Li is different on two electrodes due to the stripping/deposition process to give a voltage different, then a slowly decreasing of voltage means a leakage through the electrolyte. Authors should give a better comment on this question.

****We thank the reviewer's good comment here. We apologize for the misunderstanding of the reviewer's intention about this question in our first round response. We carefully considered the polarization effect in our reported solid state three-dimensional (3D) composite anode. The polarization of the electrode could be divided into electrochemical polarization (EP), concentration polarization (CP) and ohmic polarization (OP). Usually, what kind of polarization played the dominate role depends**

on the polarization current density (net current density, denoted as i) and exchange current density (denoted as i_0). If i is much smaller than i_0 , we only take EP into consideration for convenience. With i increasing, the CP could not be ignored. For the PEO-DLSL symmetric cell, the overpotential was about ~ 100 mV when cycled at high current densities ($\geq 500 \mu\text{A cm}^{-2}$), indicating a noteworthy CP. In this case, when i was turned to zero, the depolarization of EP occurred at once. However, Li^+ ion transportation and diffusion rate at the electrode/electrolyte interface and in the electrolyte is order of magnitudes smaller than the electrochemical reaction rate. Therefore, the depolarization of CP required time to recover homogeneous Li^+ concentration distribution and achieve steady state due to the unsatisfied Li^+ ion conductivity in the solid-state cell, and thus the voltages slowly decreased to zero when the polarizations were switched off. Furthermore, the Li^+ ion diffusion pathway in our 3D hierarchical PEO-DLSL electrode is much longer than that at the planar Li-PEO electrolyte interface. Hence, the slowly decreasing of voltage is more obvious for PEO-DLSL than Li foil. In addition, the similar slow depolarization process was also observed in a previous work with 3D anode design (Figure 3c, Liu, Y et al. *Sci. Adv.* **3**, eaa0713, 2017). *We have included the corresponding discussion in the revised manuscript (Page 17, line 5~14 in the revised manuscript).*

7. For the Li stripping/plating process, could authors explain why Li will grow back into the diatomite templates? Naturally, the system would find the lowest resistance paths for both electrons and Li-ions to deposit Li atoms which are most likely next to the interface between solid electrolyte and negative electrode. With a soft SPE, why Li has to deposit along the hierarchical structure? What makes Li would like to deposit at some place far away from the interface, especially the Li diffusion path is much longer?

**** We thank the reviewer's very nice comment here. As the reviewer pointed out that Li atoms would be deposited at where the sufficient Li^+ ions and electrons are both provided. Usually, in the traditional electrode, the electronic conductivity is much higher than that of Li^+ ions and thus the deposition of Li is likely to be appeared at the interface between solid electrolyte and the anode. But in our fabricated composite electrode, the Li^+ ion conductivity is dominant to make the deposition of Li occurring in the electrode framework. As shown in Figure R8, the Li^+ ions can be conducted in the PEO-SPE coated $\text{Li}_{4.4}\text{Si}$ -PEO framework with high conductivity but the electrons are hard to be conducted in the framework due to its low electronic conductivity. The electron transfer is likely to be conducted in the metallic Li. Therefore, Li would prefer to deposit back into the $\text{Li}_{4.4}\text{Si}$ -PEO framework (denoted in red dash circle) along the hierarchical structure rather than the electrolyte/electrode interface in our fabricated composite anode. *We have included the explanation in the discussion part in the revised manuscript and supporting information (Page 22, line 1-13; Page 20, Supplementary Figure 17 in supporting information).***

Figure R8. Schematic illustration of Li deposition mechanism in our fabricated hierarchical composite anode.

The answer seems reasonable but rather intuitively. I agree that Li will deposit firstly on the two-phase boundary (the red dash circle). However, Li is a solid metal instead of liquid which has a very low Li diffusion of $5 \times 10^{-11} \text{ cm}^2/\text{s}$. This means Li will only deposit as a very thin film on the surface of hierarchical composite and grow long the hierarchical composite to the interface, instead of fill up the gaps between the hierarchical composite. Once the Li on the surface of hierarchical composite grows to the interface of electrode/electrolyte, Li will only deposit (accumulate) at the interface because the shortest distance for Li ions to travel. Again, Li is a solid metal instead of liquid. Why it has to fill up the gap faster than grow long the composite with such a low self-diffusion?

We thank the reviewer's good comment here. We carefully reconsidered and agree that the schematic picture in **Figure R8 was not very accurate. First, the pores and gaps in the PEO-Li_{4.4}Si framework (PEO-SPEs coated Li_{4.4}Si, denoted as PEO-Li_{4.4}Si) was in the scale between submicrometers to micrometers, narrower than that shown in **Figure R8**, which was confirmed by the SEM images in Supplementary Figure 7 and Supplementary Figure 11b in the revised manuscript. Second, due to the low self-diffusion coefficient of Li, "fresh" Li would always firstly deposit at the bottom Li layer/PEO-Li_{4.4}Si interface (interface I, **Figure R8-1a**), where Li⁺ ion could travel shortest distance through the PEO-Li_{4.4}Si framework to gain the electron. It's noteworthy that the aforementioned "travel distance" kept the same in the further Li deposition for interface I. Therefore, the Li⁺ ion could be conducted to the interface I and deposited during the whole plating process, which would lead to further Li plating. Therefore, Li firstly arises at the interface I and then grows along the horizontal direction to fill the gap. At the same time, the deposited Li could conduct e⁻ to upper adjacent framework particles surface, generating a new electron/Li⁺ interface (interface II) and resulting in the rise of plated Li (**Figure R8-1b**). In summary, Li plating occurs at two orientations to fill the whole framework up (**Figure R8-1c**). The proposed Li plating mechanism is also consistent with the previously reported Li plating in 3D solid electrolyte framework (Figure 3 and Figure 4, Yang, C. et al. *Proc. Natl. Acad. Sci. USA* 115, 3770-3775, 2018). We have included the corresponding discussion and correction in the revised manuscript and supporting information (Page 22, line 5, line 11~18 and revised Figure 1 in the manuscript; revised Supplementary Figure 17 in the supporting information).

Figure R8-1. Schematic illustration of Li deposition mechanism in our fabricated hierarchical composite anode. **a**, Li deposition at interface I. **b**, Generation of interface II. **c**, Li plating occurs at both horizontal and vertical direction to fill the whole framework.

REVIEWERS' COMMENTS:

Reviewer #3 (Remarks to the Author):

Thank the authors for the detail answers for my questions. My final comment to previous comment 6 is that authors should consider that a depolarization EP will immediately give a you a negative volatge instead of slow decrease of voltage due to the back flow of ions. CP would only happen when Li was take out from Si.

Reviewer has no more comment for the rest parts of the manuscript. It can be accepted for publication.

“Diatomite derived hierarchical hybrid anode for high performance all solid state Li metal batteries”

Reviewer #3 (Remarks to the Author):

Thank the authors for the detail answers for my questions. My final comment to previous comment 6 is that authors should consider that a depolarization EP will immediately give a you a negative voltage instead of slow decrease of voltage due to the back flow of ions. CP would only happen when Li was taken out from Si.

Reviewer has no more comment for the rest parts of the manuscript. It can be accepted for publication.

With the great comments from the reviewer, the quality of our work is significantly improved. We thank the reviewer for strong support on the publication of this work. In our opinion, CP could also happen at the Li/PEO-SPE interface, where Li/Li^+ redox reaction would occur. As shown in the schematic figure below, abundant of Li^+ generated at the “stripping” interface at a high net current density. Then R-Li^+ was conducted to the “plating” interface and consumed due to deposition. Due to the limited Li^+ conductivity of the PEO-SPE, the Li^+ transportation speed could not match the Li^+ generation and consumption speed, leading to the transparency of Li^+ concentration and the corresponding CP.

Manuscript ID: NCOMMS-18-35402-D

“Diatomite derived hierarchical hybrid anode for high performance all-solid-state lithium metal batteries”

Reviewer #1 (Remarks to the Author):

The authors have addressed my concerns and questions, I would like to suggest "Acceptance" for this revised manuscript.

We thank the reviewer for strong support on the publication of this work.

Reviewer #2 (Remarks to the Author):

The authors have revised the manuscript according to three reviewers' comments and suggestions in this revision. In this reviewer's opinion, this revised version can be accepted.

We thank the reviewer for strong support on the publication of this work.

Reviewer #3 (Remarks to the Author):

Reviewer has no more comment for the rest parts of the manuscript. It can be accepted for publication.

We thank the reviewer for strong support on the publication of this work.